



# Non-asymptotic distributions of water extremes: Superlative or superfluous?

Francesco Serinaldi[1,2], Federico Lombardo[3], and Chris G. Kilsby[1,2]

[1]School of Engineering, Newcastle University, Newcastle Upon Tyne, NE1 7RU, UK
[2]Willis Research Network, 51 Lime St., London, EC3M 7DQ, UK
[3]Corpo Nazionale dei Vigili del Fuoco, Ministero dell'Interno, Piazza del Viminale, 1, Rome 00184, Italy

**Correspondence:** Francesco Serinaldi (francesco.serinaldi@ncl.ac.uk)

**Abstract.** Non-asymptotic ($\mathcal{NA}$) probability distributions of block maxima (BM) have been proposed as an alternative to asymptotic distributions of BM derived by classic extreme value theory (EVT). Their advantage should be the inclusion of moderate quantiles as well as extremes in the inference procedures. This would increase the amount of used information and reduce the uncertainty characterizing the inference based on short samples of BM or peaks over high threshold. In this study, we show that $\mathcal{NA}$ distributions of BM suffer from two main drawbacks that make them of little usefulness for practical applications. Firstly, unlike classic EVT distributions, $\mathcal{NA}$ models of BM imply the preliminary definition of their conditional parent distributions, which explicitly appears in their expression. However, when such conditional parent distributions are known or estimated also the unconditional parent distribution is readily available, and the corresponding $\mathcal{NA}$ distribution of BM is no longer needed, as it is just an approximation of the upper tail of the parent. Secondly, when declustering procedures are used to remove autocorrelation characterizing hydro-climatic records, $\mathcal{NA}$ distributions of BM devised for independent data are strongly biased even if the original process exhibits low/moderate autocorrelation. On the other hand, $\mathcal{NA}$ distributions of BM accounting for autocorrelation are less biased but still of little practical usefulness. Such conclusions are supported by theoretical arguments, Monte Carlo simulations, and re-analysis of sea level data.

## 1 Introduction

In the last decades, the statistical analysis of hydro-climatic extremes has mainly relied on theoretical results and models developed by a branch of statistics called extreme value theory (EVT) (Fisher and Tippett, 1928; Von Mises, 1936; Gnedenko, 1943; Jenkinson, 1955; Gumbel, 1958; Balkema and de Haan, 1974; Pickands III, 1975; Leadbetter, 1983; Smith, 1984; Davison and Smith, 1990; Coles, 2001; Beirlant et al., 2006; Salvadori et al., 2007). EVT describes the extremal behavior of observed phenomena by asymptotic probability distributions that are valid under certain assumptions about the parent process, such as large sample sizes $n$ (i.e. $n \to \infty$ to guarantee asymptotic convergence), independence, and distributional identity. However, hydro-climatic records are commonly quite short and hardly ever behave as independent and identically distributed random variables. More often, hydro-climatic processes result from combinations of heterogeneous physical processes (e.g., Morrison and Smith, 2002; Smith et al., 2011, 2018), they exhibit autocorrelation (e.g., Kantelhardt et al., 2006; Wang et al., 2007; Serinaldi, 2010; Labat et al., 2011; Papalexiou et al., 2011; Serinaldi and Kilsby, 2016b; Lombardo et al., 2017; Iliopoulou et al.,





2018; Markonis et al., 2018; Serinaldi and Kilsby, 2018; Serinaldi et al., 2018, and references therein), and their behavior is better described by stochastic processes incorporating such properties (e.g., Serinaldi and Kilsby, 2014a; Serinaldi and Lombardo, 2017a, b; Papalexiou, 2018; Koutsoyiannis, 2020; Papalexiou and Serinaldi, 2020; Papalexiou et al., 2021; Papalexiou, 2022; Serinaldi et al., 2022a, and references therein).

As a consequence, the lack of fulfillment of EVT assumptions affects the analysis of block maxima (BM) or over thresh-
old (OT) values, as the BM and OT sample selection generally yields short sample sizes and does not remove the effects of autocorrelation and possible heterogeneity of the generating mechanisms (see e.g., Koutsoyiannis, 2004; Iliopoulou and Koutsoyiannis, 2019; Serinaldi et al., 2020b). Research in EVT has addressed these issues to some extent for the case of asymptotic and sub-/pre-asymptotic methods for BM and OT processes (see Serinaldi et al. (2020b) and references therein for an overview).

On the other hand, a parallel literature has focused on non-asymptotic ($\mathcal{NA}$) approaches for BM, attempting to use as many observations as possible to infer the distribution of the largest values. $\mathcal{NA}$ distributions of BM include Todorovic distributions and their special cases (e.g., Todorovic, 1970; Todorovic and Zelenhasic, 1970; Lombardo et al., 2019), the so-called Metastatistical Extreme Value (MEV) distributions and their variants, such as Simplified MEV (SMEV; Marani and Ignaccolo, 2015; Zorzetto et al., 2016; De Michele and Avanzi, 2018; Marra et al., 2018; De Michele, 2019; Marra et al., 2019; Hosseini et al.,
2020; Miniussi et al., 2020; Zorzetto and Marani, 2020).

Serinaldi et al. (2020b) explained the conceptual and analytical relationships among the above-mentioned $\mathcal{NA}$ distributions of BM in the context of compound distributions of order statistics, and introduced compound beta binomial distributions ($\beta\mathcal{B}C$) of BM of processes with stationary autocorrelation structure. $\beta\mathcal{B}C$ distributions allow one to avoid declustering procedures required for instance by (S)MEV to obtain samples fulfilling the assumption of independence.

However, while $\beta\mathcal{B}C$ distributions allow for a correct interpretation of $\mathcal{NA}$ models of BM and their connections with their parent distributions, Serinaldi et al. (2020b) did not set out to comprehensively explore the usefulness or otherwise of $\mathcal{NA}$ models of BM in practical analysis. In this study, we further explore and discuss the extent of redundancy of such models with respect to their parent distributions as well as the actual lack of effectiveness of declustering procedures in the context of $\mathcal{NA}$-based analysis.

This paper falls in the class of so-called neutral (independent) validation/falsification studies (see e.g., Popper, 1959; Boulesteix et al., 2018, and references therein) aiming at independently checking the theoretical consistency in statistical methods applied in analysis of hydro-climatic data (Lombardo et al., 2012, 2014, 2017, 2019; Serinaldi and Kilsby, 2016a; Serinaldi et al., 2015, 2018, 2020a, b, 2022b). We put emphasis on the common but misleading habit of seeking for confirmation by iterating the application of a given method to observed data whose generating process is inherently unknown. In
fact, if a method is technically flawed, its output will always be consistent across applications but systematically incorrect. On the contrary, genuine neutral analysis calls into question the theory behind a method/model and checks it analytically and/or against challenging controlled conditions via suitable Monte Carlo simulations.

This study is organized as follows. In Section 2, we briefly review the main $\mathcal{NA}$ distributions of BM proposed in the literature and their relationship with the corresponding distributions of the parent process. Section 3 recalls the rationale of performing





extreme value analysis and explains why $\mathcal{NA}$ models of BM are conceptually redundant in this context. These aspects are further discussed in Section 4 using simple Monte Carlo simulations, and re-analyzing sea level data previously studied in the literature. Monte Carlo experiments in Section 5 investigate the performance of some $\mathcal{NA}$ models of BM under independence and serial dependence, the effectiveness of declustering methods proposed to deal with autocorrelated time series, and the reliability of some results previously reported in the litarture. In Section 6, the problems concerning the use of $\mathcal{NA}$ models of

BM for practical applications are placed in the wider context of a questionable approach to applied statistics in hydro-climatic studies. Conclusions are summarized in Section 7.

## 2 Overview of $\mathcal{NA}$ distributions of BM

To support our discussion, we firstly recall some basic theoretical results, referring to Serinaldi et al. (2020b) and references therein for more details about the analytical derivation of equations reported below. Under the assumption of identical proba-

bility distribution, BM are the largest order statistics (David and Nagaraja, 2004, p. 1) of a sequence of $m$ random variables $Z_1, ..., Z_m$ with the same cumulative distribution function (cdf) $F_Z(z)$. If these variables are also independent, the cdf of BM, $Y$, in random samples of finite size $m$ is

$$
\begin{aligned}
F_Y(z) &= \sum_{i=m}^{m} \binom{m}{i} F_Z^i(z)[1 - F_Z(z)]^{m-i} = F_Z^m(z) \\
&= 1 - F_{\mathcal{B}}(m-1; m, F_Z(z)) \\
&= F_{\mathcal{B}}(0; m, 1 - F_Z(z)) \\
&= F_{\beta}(F_Z(z); m, 1),
\end{aligned}
\tag{1}
$$

where $F_{\mathcal{B}}$ and $F_{\beta}$ are the binomial and beta cdf's, respectively. Under the assumption of serial dependence, the distribution of BM in finite-size blocks is unknown as it depends on the $m$-dimensional joint distribution of the $m$ variables forming a

block (Todorovic, 1970; Todorovic and Zelenhasic, 1970). Closed-form solutions do exist for the case of Markovian processes, whereby the joint distribution is bivariate (Lombardo et al., 2019). For high-order dependence structures, the $\mathcal{NA}$ distribution of BM can be approximated by beta-binomial distribution $\beta\mathcal{B}$ (Serinaldi et al., 2020b, Section 2.2)

$$
\begin{aligned}
F_Y(z) &= \frac{\mathrm{B}(\alpha(z), m + \beta(z))}{\mathrm{B}(\alpha(z), \beta(z))} \\
&= F_{\beta\mathcal{B}}(0; m, 1 - F_Z(z), \rho_{\beta\mathcal{B}}(F_Z(z), \boldsymbol{\rho})),
\end{aligned}
\tag{2}
$$

where $F_{\beta\mathcal{B}}$ is the $\beta\mathcal{B}$ cdf, $\mathrm{B}(\cdot, \cdot)$ is the complete beta function (Arnold et al., 1992, pp. 12-13), $\rho_{\beta\mathcal{B}}(z)$ is known as the 'intra-class' or 'intra-cluster' correlation, which depends on $F_Z(z)$ and the autocorrelation function (ACF) of the parent process

$\{Z_i\}_{i=1}^{m}$, denoted as $\boldsymbol{\rho}$. When the parent process $Z_i$ is serially uncorrelated ($\rho_{\beta\mathcal{B}} = 0$), Eq. 2 yields Eq. 1 as a particular case.





The process $Z$ is named 'parent' as it is the stochastic process whose distribution $F_Z$ appears in the expression of the distribution of BM $F_Y$, and it could have a no strict physical meaning. For example, the parent process used to build the distribution of BM for precipitation or stream flow sampled at a given time scale (e.g., daily) could be the process of observations over any threshold guaranteeing the selection of at least one observation per block. Therefore, inter-arrival times of the observations $z$ are always smaller than or equal to the $m$ time steps corresponding to the block size. As a limiting case, $Z$ can obviously be the complete stream flow or rainfall process sampled at the finest time scale (e.g., daily).

As discussed in more depth in the next sections, every distribution of BM (asymptotic or non-asymptotic) provide just an approximation of the upper tail of the distribution of the parent process. Eqs. 1 and 2 indicate that two parent processes can have the exact marginal distribution, but the expression of the corresponding $\mathcal{NA}$ model of BM approximating the upper tail of $F_Z$ might be different according to the presence or absence of serial dependence. In other words, serial dependence influences the patterns of the observations $z$ within each block, and therefore the sequences of BM and the form of their $\mathcal{NA}$ distribution $F_Y$. On the other hand, $F_Z$ is unaffected by serial dependence as it describes the distribution of $Z$, which do not imply any operation (aggregation, average, or BM selection) over a time window (block).

The assumption of intra-/inter-block distributional identity can be relaxed by resorting to the concept of mixed/compound distributions, which integrate (average) over the parameter space of the parent distribution, under the assumption that these parameters can change within/between each block (Marra et al., 2019; Serinaldi et al., 2020b). For instance, such changes/fluctuations can reflect different physical generating mechanisms (e.g., convective and frontal weather systems generating storms in different seasons) or inter-block sampling uncertainty related to still unidentified physical processes, which therefore need a stochastic description. A general compact form of this class of models can be written as

$$F_Y(z) = \sum_{l=0}^{\infty} \int_{\Omega_{\boldsymbol{\theta}}} G_l(z;\boldsymbol{\theta})g(l,\boldsymbol{\theta})\mathrm{d}\boldsymbol{\theta} = \mathbb{E}\left[G_l(z;\boldsymbol{\theta})\right], \tag{3}$$

where $G_l(z;\boldsymbol{\theta}) = \mathbb{P}[Z_1 \leq z \wedge Z_2 \leq z \wedge ... \wedge Z_l \leq z | L = l, \boldsymbol{\Theta} = \boldsymbol{\theta}]$ is the joint distribution of the parent process accounting for intra-block dependence, $\Omega_{\boldsymbol{\theta}}$ is the state space of parameter vector $\boldsymbol{\theta}$, and $\mathbb{E}[\cdot]$ is the expectation operator. $G_l$ is integrated (averaged) over the number of observations $L$ in each block of size $m$ and the parameters $\boldsymbol{\Theta}$, which are treated as random variables with joint probability density function (pdf) $g(l,\boldsymbol{\theta})$. Equation 3 is a generalization of Todorovic distributions incorporating possible inter-block fluctuations of parameters of the joint distribution of parent process $Z$.

Since high-dimensional joint distributions $G_l$ are difficult to handle and fit, the general model in Eq. 3 can be approximated by a compound version of the $\beta\mathcal{B}$ distribution in Eq. 2 for high-order dependence structures, resulting in the following compound $\beta\mathcal{B}$ model ($\beta\mathcal{B}$C) (Serinaldi et al., 2020b, Section 5.2)

$$F_Y(z) \cong F_{\beta\mathcal{B}\mathrm{C}}(z) := \sum_{l=0}^{\infty} \int_{\Omega_{\boldsymbol{\rho}}} \int_{\Omega_{\boldsymbol{\theta}}} F_{\beta\mathcal{B}}(0;l,1-F_Z(z;\boldsymbol{\theta}),\rho_{\beta\mathcal{B}}(F_Z(z;\boldsymbol{\theta}),\boldsymbol{\rho}))g(l,\boldsymbol{\rho},\boldsymbol{\theta})\mathrm{d}\boldsymbol{\rho}\mathrm{d}\boldsymbol{\theta}$$

$$= \mathbb{E}\left[F_{\beta\mathcal{B}}(0;l,1-F_Z(z;\boldsymbol{\theta}),\rho_{\beta\mathcal{B}}(F_Z(z;\boldsymbol{\theta}),\boldsymbol{\rho}))\right], \tag{4}$$





where $F_{\beta\mathcal{B}C}$ is the $\beta\mathcal{B}C$ cdf, $\boldsymbol{\rho}$ is correlation matrix of the parent process $Z$, and $\Omega_{\boldsymbol{\rho}}$ is its state space. Under the assumption of independence ($\boldsymbol{\rho} = 0$), the $\beta\mathcal{B}$ distribution reduces to a binomial distribution (which can also be written in the form of a beta distribution), and Eq. 4 yields MEV models as special cases

$$
\begin{aligned}
F_Y(z) &= \sum_{l=0}^{\infty} \int_{\Omega_{\boldsymbol{\theta}}} F_{\mathcal{B}}(0; l, 1 - F_Z(z; \boldsymbol{\theta})) g(l, \boldsymbol{\theta}) \mathrm{d}\boldsymbol{\theta} \\
&= \sum_{l=0}^{\infty} \int_{\Omega_{\boldsymbol{\theta}}} F_{\beta}(F_Z(z; \boldsymbol{\theta}); l, 1) g(l, \boldsymbol{\theta}) \mathrm{d}\boldsymbol{\theta} \\
&= \sum_{l=0}^{\infty} \int_{\Omega_{\boldsymbol{\theta}}} F_Z^l(z; \boldsymbol{\theta}) g(l, \boldsymbol{\theta}) \mathrm{d}\boldsymbol{\theta}.
\end{aligned}
\tag{5}
$$

Analogously to Eqs. 1 and 2, Eqs. 4 and 5 approximate the upper tail of the distribution of the parent process $Z$

$$
F_Z(z) = \int_{\Omega_{\boldsymbol{\theta}}} F_Z(z; \boldsymbol{\theta}) g(\boldsymbol{\theta}) \mathrm{d}\boldsymbol{\theta}
\tag{6}
$$

which is itself a compound distribution (averaged over the parameter space) and should not be confused with the conditional distributions $F_Z(z; \boldsymbol{\theta})$, which depend on the parameters. $F_Z$ in Eq. 1 is also unaffected by serial correlation, which in turn changes the form of the corresponding $\mathcal{NA}$ distribution $F_Y$ of BM. As mentioned above, we can have two parent processes with identical $F_Z$ and different $F_Y$ depending on the presence or absence of serial dependence. Eqs. 4 and 5 are quite general and account not only for inter-block fluctuations via $g(\boldsymbol{\theta})$, but also intra-block variability (such as different physical generating mechanisms and/or seasonal fluctuation acting at intra-block scale) assuming that the conditional distributions $F_Z(z; \boldsymbol{\theta})$ are compound/mixed, that is

$$
F_Z(z; \boldsymbol{\theta}) = \int_{\Omega_{\boldsymbol{\vartheta}}} F_Z(z; \boldsymbol{\vartheta}, \boldsymbol{\theta}) g(\boldsymbol{\vartheta}; \boldsymbol{\theta}) \mathrm{d}\boldsymbol{\vartheta},
\tag{7}
$$

where $g(\boldsymbol{\vartheta}; \boldsymbol{\theta})$ describes the intra-block variability of $\boldsymbol{\vartheta}$ (e.g., seasonal fluctuations or intra-annual weather systems' switching) conditioned on the inter-block status (e.g., El Niño/La Niña conditions spanning one or more years). Of course, $g(\boldsymbol{\vartheta}; \boldsymbol{\theta})$ reduces to $g(\boldsymbol{\vartheta})$ if the intra-block fluctuations are assumed to be independent of inter-annual fluctuations. A typical example is the common assumption of year-to-year invariant seasonal patterns.

In the next sections, models in Eqs. 1, 2, 4 and 5 are compared with the corresponding parent distributions. We stress that the models in Eqs. 4 and 5 must be compared with the corresponding compound parent distribution in Eq. 6, which accounts for the same intra-/inter-block variability. It is worth noting that the following discussion is fully general and valid for any $\mathcal{NA}$ model of BM requiring the preliminary knowledge/definition of $F_Z(z; \boldsymbol{\theta})$ and its use in the expression of $F_Y$. Hereinafter, the terms '$\mathcal{NA}$ model/distribution of BM' and '$\mathcal{NA}$ model/distribution' are used interchangeably to denote the same class of models.





# 3 Modeling extreme values: asking 'why' before looking for 'how'

Asymptotic distributions provided by EVT are the limit distributions of $\mathcal{NA}$ models under some assumptions concerning the nature of the marginal distribution and dependence structure of the parent process $Z$. In particular, it is well known that the

Generalized Extreme Value (GEV) and Generalized Pareto (GP) distributions are the general asymptotes of the distributions of BM and peaks over thresholds (POT), respectively, under independence (or certain types of weak dependence) and distributional identity (see e.g., Leadbetter et al., 1983; Coles, 2001). Therefore, EVT models are fairly general and relatively easy to apply mainly because they do not require a precise knowledge of $F_Z$ (Leadbetter et al., 1983, p. 4), which instead explicitly appears in the expression of any $\mathcal{NA}$ model. This aspect has already been stressed in standard handbooks of applied statistics

such as Mood et al. (1974, p. 258), who stated (using our notation and setting $L = m$) "*One might wonder why we should be interested in an asymptotic distribution of $Y$ when the exact distribution, which is given by $F_Y(z) = F_Z^m(z)$, where $F_Z$ is the c.d.f.* [cumulative distribution function] *sampled from, is known. The hope is that we will find an asymptotic distribution which does not depend on the sampled c.d.f. $F_Z$. We recall that the central-limit theorem gave an asymptotic distribution for $\bar{Z}$* [sample mean] *which did not depend on the sampled distribution even though the exact distribution of $\bar{Z}$ could be found.*"

Bearing in mind that $Z$ and $Y$ are two different processes (Serinaldi et al., 2020b, Sect 3.2), the usefulness and widespread application of asymptotic EVT models of BM and POT stems from the fact that such distributions approximate (converge to) the upper tail of the distribution of the parent process $Z$ without needing to know $F_Z$ (under the above-mentioned assumptions) and just requiring a limited amount of information (i.e., BM and/or POT observations) instead of complete time series. This is paramount in practical applications as it allows the use of (i) a couple of general distributions (GEV and GP) supported by a

theory that clearly identifies the range of validity of such models, and (ii) data that are more easy to collect and widely available worldwide compared to complete time series. For example, meteorological services provide most of the historical information on rainfall in terms of annual maximum values for specified durations to be used in the so-called intensity-duration-frequency (IDF) analysis. In these cases, we do not know $F_Z$ and we cannot fit it either, as the data representing the whole rainfall process, and therefore $F_Z$, are not available. However, EVT states for instance that the GEV distribution asymptotically approximates

the upper tail of $F_Z$ independently of the form of $F_Z$ (under certain constraints) based on theoretical results concerning the asymptotic behavior of $F_Y = F_Z^m$. EVT distributions independent of the form of $F_Z$ are also useful when observations of $Z$ are available, but defining a reliable model for $F_Z$ is too difficult due to complexity of the hydro-climatic process of interest and its generating mechanisms.

Unlike asymptotic models, $\mathcal{NA}$ distributions require the preliminary knowledge/fit of $F_Z$, which explicitly appears in their

expression. However, if we already know $F_Z$ (or we have a good estimate of it), we no longer need any $\mathcal{NA}$ distribution of BM, as the latter provides just an approximation of the upper tail of the known/fitted $F_Z$. We do not even need any asymptotic model, and more generally any model of BM or POT, as these are just processes extracted from the parent process $Z$ whose distribution $F_Z$ already describes the whole state space, including the extreme values. The use of extreme value distributions makes sense if and only if we do not have enough information on $F_Z$. Otherwise, the latter provides all information needed to





make statements about any quantile. In this context, $F_Z^m$ only plays a functional/intermediate role in theoretical derivations to move from $F_Z$ to general asymptotic distributions independent of $F_Z$, to be used when $F_Z$ is not available.

The same remarks hold true for any compound $\mathcal{NA}$ model such as $\beta\mathcal{BC}$ and its special cases. In fact, these models require the preliminary inference of $F_Z$ to derive distributions (compound versions of $F_Z^m$) that only approximate the upper tail of the previously estimated $F_Z$. It is easy to understand that such a procedure makes little sense in practical applications: why should
one search for an approximation of the upper tail of a distribution that is already known or fitted? The use of compound $\mathcal{NA}$ models is not even justified by their mixing nature, which allows for averaging inter-block fluctuations of parameters. In fact, as further discussed below, such a mixing procedure can directly be applied to $F_Z$, thus obtaining a compound distribution of the parent process $Z$ that can readily be used to make statements on any quantile, avoiding unnecessary $\mathcal{NA}$ approximations of the upper tail. This explains why $\mathcal{NA}$ have not received much attention and why the recently proposed compound $\mathcal{NA}$ models are
of little practical usefulness, if any. Their usefulness is mainly theoretical, as they help explain the inherent differences between parent processes $Z$ and BM processes $Y$, thus avoiding misconceptions and misinterpretation of different model outputs (see Serinaldi et al., 2020b).

## 4   Do we need $\mathcal{NA}$ distributions of BM in practical applications? Investigating circular reasoning and redundancy

Albeit the concepts discussed in Section 3 should be well-known and self-evident, they seem to be systematically neglected
in hydro-climatic literature dealing with $\mathcal{NA}$ models. Therefore, this section reports further discussion using some simple examples and real-world data re-analysis to highlight the relationship between $\mathcal{NA}$ models and the embedded distribution $F_Z$, thus showing concretely how the former provide just a redundant approximation of the upper tail of the latter.

### 4.1   Estimation of $\mathcal{T}$-year events: recalling basic concepts to avoid inconsistencies

The first example is freely inspired by the work of Mushtaq et al. (2022), who searched for an approach to select the most
suitable distribution $F_Z$ of ordinary stream flow peaks (i.e., the parent process $Z$) between Gamma and Log-normal to be used to build MEV distributions $F_Y$ for annual maxima (AM, i.e., the BM process $Y$). Here, we focus on the very primary logical contradiction (circular reasoning) of attempting to find a distribution $F_Z$ to build $F_Y$ as a function of $F_Z$ to approximate the tail of $F_Z$ itself, which is already known exactly. In this respect, to keep the discussion as simple and focused as possible but without loss of generality, we do not use compound models but assume that the parent process is independent and identically
distributed, following a Gamma distribution. Compound models and the issues related to some MEV technicalities (such as the declustering method used to obtain apparently independent ordinary events) will be discussed in the second example. Concerning the first example, we firstly discuss the above mentioned contradiction (circular reasoning) from a conceptual perspective and then provide visual illustration by Monte Carlo simulations.



### 4.1.1 The logic behind the estimation of return levels and the role of $F_Z$ and $F_Y$

For the sake of illustration, let us suppose we have a hypothetical stream flow process sampled at daily time scale, and we are interested in estimating a flow value exceeded on average every $\mathcal{T}$ years, i.e., the so-called $\mathcal{T}$-year return level corresponding to $\mathcal{T}$-year return period (see e.g., Eichner et al., 2006; Serinaldi, 2015; Volpi et al., 2015, and references therein). Under the ideal situation that infinitely long records are available and therefore $F_Z$ and $F_Y$ are known exactly, one can use the distribution of the parent process $F_Z$ and determine the $\mathcal{T}$-year return level as the quantile $z_p$ that is exceeded with probability $p = 1/(365\mathcal{T})$, i.e.,

the value exceeded on average once in $\mathcal{T}$ years $= 365\mathcal{T}$ days (leaving aside leap years). Since $z_p$ is a quantile of the distribution $F_Z$, which describes the parent process at its finest available resolution (here, daily), it is unaffected by possible autocorrelation and clustering of $\mathcal{T}$-year events (see Bunde et al., 2004, 2005; Serinaldi et al., 2020b, for an in-depth discussion). Note that this is the definition applied in the literature to compute the exact $\mathcal{T}$-year return level used to assess the accuracy of $\mathcal{NA}$ models (see e.g., Marani and Ignaccolo, 2015; Marra et al., 2018).

However, real world records rarely span more than a few decades, and data are not enough to obtain $F_Z$ (and $F_Y$) and determine directly the $\mathcal{T}$-year return level for high values of $\mathcal{T}$ such as 100 or 1000 years. Therefore, an alternative approach is based on the distribution of AM, i.e., BM within relatively short intervals (i.e., 365 days). Of course, a virtually infinite sequence of BM defines their exact distribution. Such a distribution allows an approximate estimation of the $\mathcal{T}$-year return level as the quantile that is exceeded with probability $1/\mathcal{T}$ because one year is the finest time scale of AM. In other words, $F_Y$

cannot provide information about events occurring more often than once in $m$ days (e.g., once per year for AM), as this is the finest sampling frequency of BM for blocks of size $m$. This estimation of $\mathcal{T}$-year based on BM involves the joint exceedance probability within each block described by the intra-block joint distribution $G_l$ (see Section 2), and therefore it is affected by autocorrelation (see Eichner et al., 2006, for a detailed discussion).

    Therefore, the distributions of AM commonly used in hydro-climatology are only approximations of the upper tail of $F_Z$,

and their estimation is justified if $F_Z$ is unknown. This can happen if (i) we have no regular records of the parent process to reliably estimate $F_Z$, or (ii) a faithful parametrization of $F_Z$ is not so easy to determine due to the difficulties to account for various characteristics of the underlying process, such as cyclo-stationarity, different physical generating mechanisms, and other possibly unknown factors. In these cases, EVT comes into play stating for instance that, under certain assumptions, the distribution of BM within relatively short intervals (e.g., 365 days) converges to one of the three asymptotic extreme value

models summarized by GEV distribution independently of the exact form of $F_Z$. Of course, the approximate/partial fulfillment of EVT assumptions affects convergence. For example, autocorrelation and lack of distributional identity slow convergence down (Koutsoyiannis, 2004; Eichner et al., 2006; Serinaldi et al., 2020b), and sometimes prevent it, resulting in degenerate models. These remarks explain why asymptotic models are so powerful tools widely applied in any discipline dealing with extreme values.



### 4.1.2 Visualizing the relationship between of $F_Z$ and $F_Y$

A simple example with graphical illustration can help better clarify the difference between $F_Z$ and $F_Y$ (see Serinaldi et al., 2020b, Section 3.2 for a formal discussion based on theoretical arguments). Let us assume that we have $365 \cdot 10^5$ observations of an independent process $Z$ following a Gamma distribution with shape and scale parameters $\kappa = \sigma = 2$, representing for instance $10^5$ years of daily records of a hypothetical stream flow process (or a generic hydro-climatic process). These data allow one to build the empirical version of $F_Z$ and $F_Y$ and the corresponding pdf's $f_Z$ and $f_Y$. In particular, Fig. 1 shows the empirical pdf's (Fig. 1a-c) and the return level plots (i.e., return level vs. return period; Fig. 1d-f) for two sub-samples of size $365 \cdot 100$ and $365 \cdot 500$ (i.e., 100 and 500 years, respectively), and the whole data set (10000 years). Figure 1 also displays the theoretical Gamma pdf and return level curves as well as empirical and theoretical 100-year quantile (vertical lines). The $\mathcal{T}$-year return levels are computed as the $(1 - \frac{1}{\mathcal{T}}) \cdot 100\%$ quantiles of the empirical cdf of AM and the $(1 - \frac{\mu}{\mathcal{T}}) \cdot 100\%$ quantiles of the theoretical and empirical cdf of the process $Z$, where $\mu = 1/365$ can be interpreted as the inter-arrival time (in years) between two records of $Z$.

For $\mathcal{T}$ greater than $\cong 20$ years, the upper tail of the empirical $F_Y$ ($f_Y$) matches that of the empirical $F_Z$ ($f_Z$). This matching and convergence to the upper tail of the theoretical $F_Z$ ($f_Z$) improve as the sample size increases. This behavior is further stressed focusing on the 100-year return levels (vertical lines in Fig. 1). It should be noted that the discrepancies between $F_Y$ and $F_Z$ for $\mathcal{T} < 20$ years do not depend on the sample size. Instead, they are related to the different nature of the processes $Y$ and $Z$, and their magnitude also depends on autocorrelation when data are correlated (see Serinaldi et al., 2020b, Sect 3.2, for a theoretical discussion). Both distributions provide very close estimates of the 100-year return level for each sample size, and the accuracy obviously improves as the sample size increases. Moreover, Fig. 1 provides an intuitive (albeit very simplified) explanation of why EVT models of BM work when $F_Z$ is not available and EVT assumptions are fulfilled.

From Fig. 1, it is evident that we do not need any model for $Y$ if we already have a model for the parent $Z$. Since $\mathcal{NA}$ distributions require the preliminary definition/fit of a model for $F_Z$, they have no practical usefulness, as the preliminarily fitted $F_Z$ already provides all the information to make statements on both ordinary and extreme events/quantiles. In this respect, defining $\mathcal{NA}$ distributions from $F_Z$ is only an unnecessary and redundant step yielding just an approximation of the embedded $F_Z$. These issues are further discussed in the next section reviewing a real-world data analysis previously reported in the literature.

### 4.2 Re-analysis of sea level data

In this section, we further illustrate the foregoing concepts by re-analyzing two sea-level time series already studied by Caruso and Marani (2022). These data refer to hourly sea-level records from the tide gauge of Hornbæk (Denmark) and Newlyn (United Kingdom), spanning 122 years (1891-2012) and 102 years (1915-2016), respectively. Data are freely available from University of Hawaii Sea Level Center (UHSLC) repository (Caldwell et al., 2015, http://uhslc.soest.hawaii.edu/data/?rquh745a/). For the sake of consistency with the original work, we removed years with less than six months of water level observations and days with less than 24 hours of data (see Caruso and Marani, 2022). This resulted in 120 and 100 years of data for Hornbæk and



**Figure 1.** Probability density functions (a-c) and return level plots (return period vs. return level; d-f) of samples of varying size ($365 \cdot \{100, 500, 10000\}$) and corresponding BM (with block size $m = 365$) drawn from a Gamma distribution. The diagrams show the relationship between parent distribution and distribution of BM along with the convergence of the upper tails of the empirical distribution toward the theoretical counterparts. The abscissa of dashed vertical lines indicates the value of the theoretical 100-year return level (gray lines) and its estimates from samples of the parent process $Z$ (blue lines) and the corresponding BM process $Y$ (red lines).





Newlyn gauges, respectively. Moreover, time series are pre-processed by filtering out the time-varying mean sea level (m.s.l.) computed using the average of daily levels for each calendar year. Thus, the filtered time series retain the contributions from
astronomical tides and storm surges.

Daily maxima are used as the basis for extreme value analysis, which is performed by three different approaches: (i) GEV distribution of AM, (ii) GP distribution of POT, and (iii) GP-based MEV of peaks over moderate threshold (i.e., the so-called ordinary events). These extreme value models assume that the underlying process is a collection of independent random variables. Since sea levels are a typical example of autocorrelated process, data are preliminarily declustered by selecting
peaks that are separated by at least 30 days, to obtain (approximately) independent samples. In more detail, Caruso and Marani (2022) adopted "*a threshold lag of 30 d, which yielded the minimum estimation error under the MEVD approach*". Therefore, declustered data are used to extract AM, and POT samples over optimal statistical thresholds (Bernardara et al., 2014). Caruso and Marani (2022) selected the GP threshold for POT by studying the stability of the GP shape parameter (Coles, 2001, p. 83), while they chose the moderate threshold of GP distributions entering MEV "*by testing different threshold values and evaluating*
*the goodness of fit of the distribution using diagnostic graphical plots*".

Before presenting results of extreme value analysis, it is worth noting that:

1. The extraction of independent data from correlated samples is referred to as 'physical declustering' (Bernardara et al., 2014). Its algorithms rely on physical properties of the process of interest (e.g., the lifetime of the weather systems generating a storm over an area) and/or properties of the occurrence process (e.g., statistics of the (inter-)arrival times of rainfall storms). In this respect, a threshold selection based on "*the minimum estimation error under the MEVD*
*approach*" does not only require iterative fitting of MEV components, but also contrasts with the rationale of physical declustering whose algorithms should be unrelated to the subsequent analysis and models involved. In other words, physical declustering should guarantee only independence of the extracted sample and not the goodness-of-fit of a specific model (GP, MEV, or anything else).

2. Goodness-of-fit concerns statistical optimization, which aims at setting a threshold that guarantees the convergence/fit of the POT sample to an extreme value model. For the GP model, such a threshold should provide "*the best compromise between the convergence of* [POT distribution toward] *a GP distribution (bias minimization) and the necessity to keep enough data for the estimation of its parameters (variance minimization)*" (Bernardara et al., 2014). In the present case, such a statistical threshold should not be required as the physical threshold was already selected to yield "*the minimum*
*estimation error under the MEVD approach*" (Caruso and Marani, 2022). In fact, for Hornbæk and Newlyn data sets, the thresholds used by Caruso and Marani (2022) (i.e., 40 and 250 cm, respectively) lead to discard approximately only the 13% of the complete declustered sample. Therefore, for the sake of comparison, we applied MEV on both the original declustered data and their over-threshold sub-samples.

For both data sets, Fig. 2 shows the time series of AM (Fig. 2a,d), POT for GP (Fig. 2b,e), and POT for MEV (Fig. 2c,f)
along with the complete sample of daily maxima. Note that the sizes of POT samples are slightly different from those reported



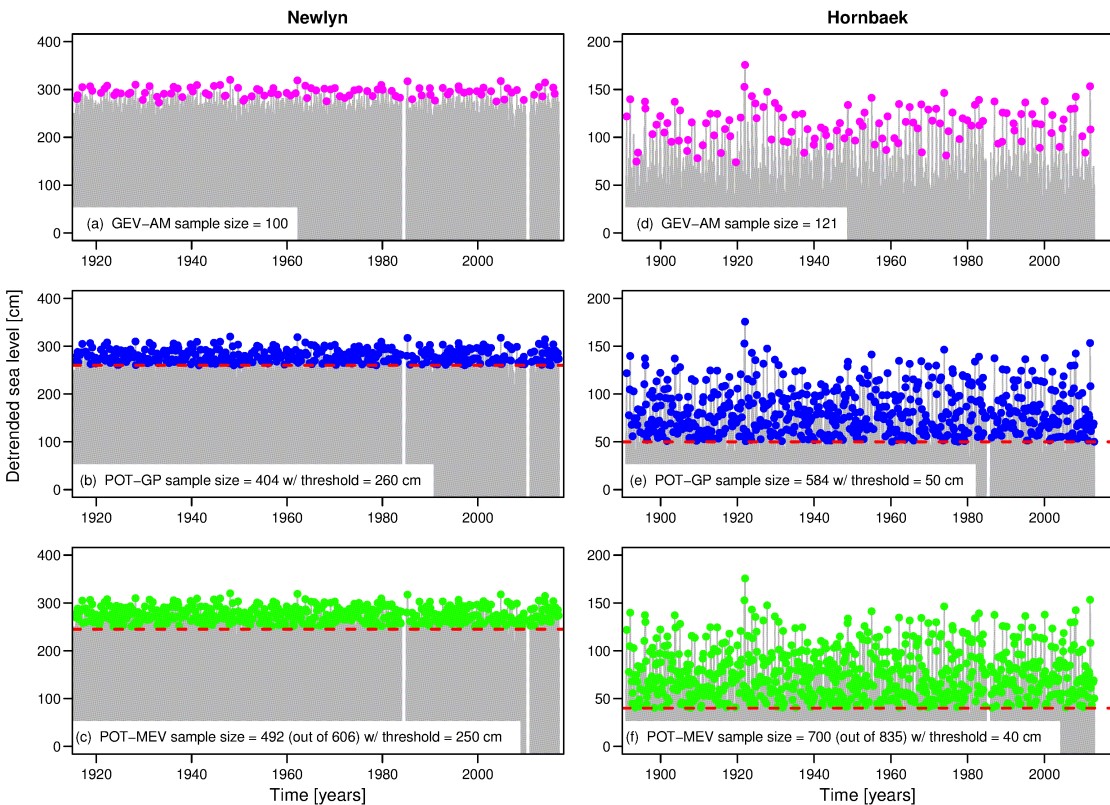

**Figure 2.** Detrended sea levels (gray line) for the gauging sites of Newlyn (UK) and Hornbæk (Denmark), and AM values for GEV analysis (a,d), POT used for GP analysis (b,e), and over threshold events used for fitting MEV and Compound parent models (c,f). 'Detrended' refers to sea level time series preliminarily filtered by removing the time-varying mean sea level.

by Caruso and Marani (2022). This is likely due to slightly different implementation of declustering algorithm, which involves some technicalities such as the treatment of not available values.

Figure 3 reports results of extreme value analysis in terms of return level plots. Figure 3a shows the empirical return level plot of AM sample used to fit the GEV distribution and that of the corresponding declustered sample used to extract AM values.

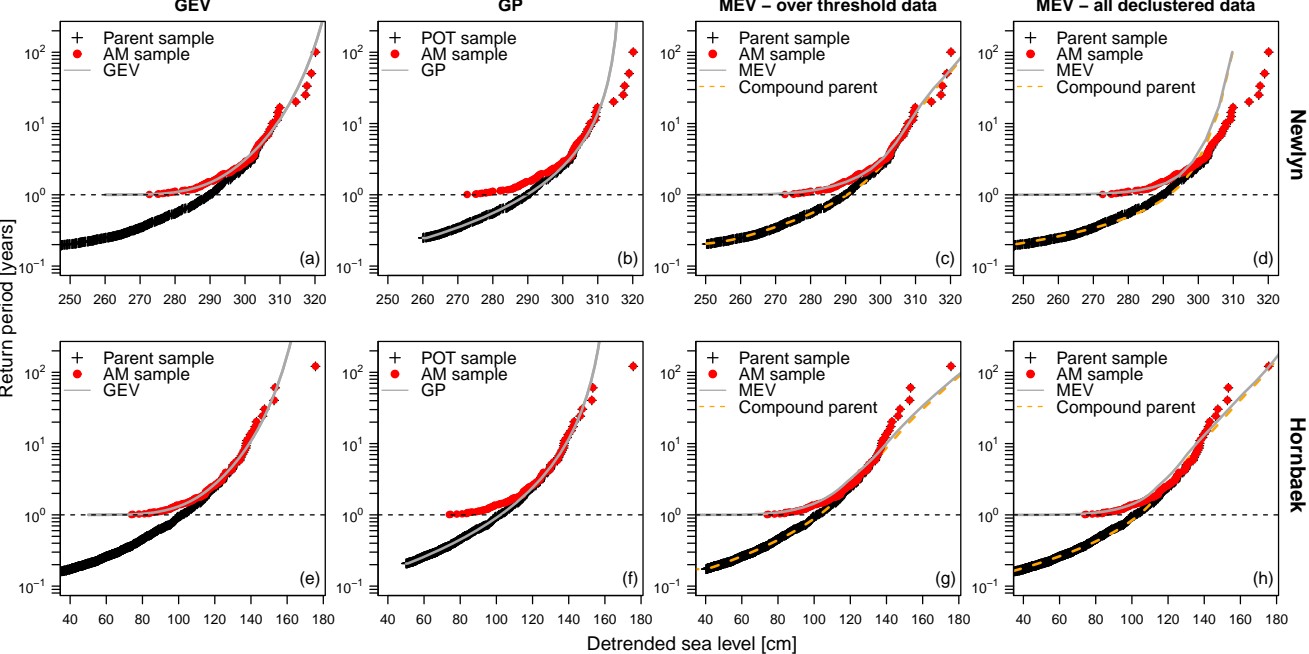

**Figure 3.** Return level diagrams (return period (in years) vs. return level) resulting from extreme value analysis of Newlyn data (a-d) and Hornbæk data (e-h). All panels report empirical return level diagrams of AM as a common reference. Panels (a,e) report empirical return level diagrams of the parent sample of declustered data along with the theoretical return level diagram of fitted GEV model. Panels (b,f) refer to POT sample and the corresponding GP model. Panels (c,g) and (d,h) show results for MEV and Compound parent distributions applied to over threshold data and complete declustered sample, respectively.

The values of return period used to build these diagrams are estimated as $\mathcal{T} = \frac{\mu}{1-F_n}$, where $F_n$ is the empirical cdf of AM or declustered sample, and $\mu$ is the average inter-arrival time between two observations of a (discrete-time) process of interest, i.e., $\mu = 1$ for AM and $\mu = \mathbb{E}[1/L]$ for the complete declustered sample, where the random variable $L$ denotes the varying number of events (or peaks) per year. Figures 3a and 3e are analogous to Fig. 1, and convey the same message but for real world-data, that is, the distribution of AM is just an approximation converging to the distribution of the parent sample for large quantiles (upper tail).

When using POT values over the threshold optimizing the GP fitting (Fig. 3b,f), we get a similar message: the distribution of AM is an approximation of the upper tail of the distribution of POT, which play a role similar to that of parent sample in $\mathcal{NA}$ models of BM. In fact, the GP-based analysis of POT does not require the subsequent derivation of the distribution of AM to make inference on return levels, as the return period (in years) of any quantile is computed as $\mathcal{T} = \frac{\hat{\mu}}{1-F_{\mathrm{GP}}}$, where $F_{\mathrm{GP}}$ is the GP cdf and $\hat{\mu}$ is the estimate of the average inter-arrival time between two POT observations. Even though this remark can seem trivial, it plays a key role to understand the redundancy of $\mathcal{NA}$ distributions.





MEV models require to preliminarily fit a model for values above a moderate threshold (or all available independent declustered data), which is our parent distribution $F_Z$, and therefore deriving the distribution of the annual maxima $F_Y$ as a function of $F_Z$. Figures 3c, 3d, 3g, and 3h show both the empirical cdf's of AM and parent sample, and their theoretical counterpart,

i.e., the GP-based MEV model and the compound GP parent. As for GEV, the MEV distribution is just an approximation of the upper tail of the fitted compound parent. However, in this case, we already have a model for the parent process, and therefore we do not need any distribution of AM, as the fitted compound $F_Z$ already provides all information required for inferential purposes. In other words, MEV cannot provide correct probability of low/moderate quantiles (as every extreme value model of BM), and it cannot add any information compared to corresponding fitted compound parent $F_Z$. Once $F_Z$ is available, any

other model of any sub-process (such as AM or POT) is less informative or redundant, at most.

Figures 3c, 3d, 3g, and 3h also show that the claimed goodness of fit of MEV models is not related to its nature of distribution of AM, but to the fact that it is a compound distribution. In fact, MEV tails match those of the corresponding compound parent distributions. When we have a good compound model $F_Z$ integrating (i.e., averaging) seasonal fluctuations and other forcing factors (such as different generating mechanisms of rainfall, storms, flood, or storm surges), the corresponding $\mathcal{NA}$ model is

no longer needed as it can at most be as accurate as the corresponding compound $F_Z$.

The use of $\mathcal{NA}$ distributions is not even justified to make inference in terms of return period and return levels. In fact, a compound $F_Z$ can be used to compute return levels in the same way as one uses GP distributions, calculating the return period as $\mathcal{T} = \frac{\hat{\mu}}{1-F_Z}$, where $\hat{\mu}$ is the estimate of the average inter-arrival time between two observations in the sample of values above a moderate threshold (as for the case in Fig. 3c and 3g) or in the complete sample of independent declustered data (as for the

case in Fig. 3d and 3h). In general, $F_Z$ does not require deriving the corresponding $\mathcal{NA}$ model for AM to make inference in terms of return period (expressed in years), in the same way as GP-based inference for POT does not require the corresponding GEV model of AM.

## 5   Smoke and mirrors on the water extremes: a matter of compound distributions, neglected dependence, and misuse of multi-model ensemble averaging

The discussion in Sections 3 and 4 was based on conceptual arguments, simplified numerical examples, and real-world data re-analysis with simple visual assessment. However, to be consistent with the scientific method, new models and methods should be validated/falsified against challenging and controlled conditions before being applied to real-world data coming from inherently unknown processes (Serinaldi et al., 2020a, 2022b). To this aim, we set up three Monte Carlo experiments. The first experiment replicates and expands the numerical simulations reported by Marra et al. (2018) with the aim to provide

independent validation and further evidence about the redundancy of $\mathcal{NA}$ models (here, MEV) when dealing with serially independent processes. The second experiment investigates the effect of autocorrelation on $\mathcal{NA}$-based analysis, evaluating the effectiveness of declustering algorithms based on threshold lags as well as the use of $\beta\mathcal{BC}$ models accounting for serial correlation without declustering. The third experiment replicates and expands some of the Monte Carlo simulations reported





by Marani and Ignaccolo (2015) to support the introduction of MEV models. In this case, the aim is to explain the apparent
discrepancies between results in Marani and Ignaccolo (2015) and those in Marra et al. (2018).

### 5.1  Monte Carlo experiment 1: serially independent processes

The first experiment consists of simulating $S = 1000$ time series of ordinary events mimicking 3, 5, 10, 20, and 50 years
of records. Each year comprises $l$ events drawn from a random variable $L$ following Gaussian distribution with mean $\mu_L \in$
$\{10, 50, 100\}$, and standard deviation $\sigma_L = 0.3\mu_L$. Marra et al. (2018) chose the range of $\mu_L$ and $\sigma_L$ based on exploratory
analysis of hourly rainfall data collected over the contiguous United States. Ordinary events are simulated from Weibull distri-
butions with shape parameter $\kappa \in \{0.8, 1.25\}$ and scale parameter $\lambda = 1$. The $\kappa$ values represent the typical range of variability
of the observed rainfall data studied by Marra et al. (2018), while constant $\lambda$ is chosen for easier interpretation of results. The
simulated time series are used to estimate the 100-year return levels. The reference 100-year return level is empirically obtained
from $10^5$ years of simulated samples, and the performance of GEV, GP, and MEV is checked in terms of multiplicative bias

$$B_k = \frac{\hat{x}_k}{x_{\text{ref}}}, \tag{8}$$

where $\hat{x}_k$ is the estimate of the target statistics (here, 100-year return level) for the $k^{\text{th}}$ Monte Carlo simulation (with $k =$
$1, ..., S$), and $x_{\text{ref}}$ is the reference (true) value.

We note that the use of a Gaussian distribution with infinite support can generate physically inconsistent negative number
of events in some years. Moreover, simulating integer values from a continuous distribution requires rounding off. In these
cases, more appropriate models for discrete random variables defined in $[0, \infty)$, such as binomial, beta-binomial, Poisson,
or geometric should be used. The reference 100-year return level can be computed as the $(1 - \frac{1}{100}) \cdot 100\%$ quantile of the
empirical cdf of AM or the $(1 - \frac{\hat{\mu}}{100}) \cdot 100\%$ quantile of the empirical cdf of the complete time series of ordinary events, where
$\hat{\mu}$ is the estimate of the average inter-arrival time (in years) between two ordinary events. For large samples, the former estimate
converges to the latter for $\mathcal{T}$ values greater than a few years (e.g., 3-5 years for independent data; see Fig. 1) or much more for
serially dependent processes (see Serinaldi et al., 2020b, Sect 3.2). In any case, the most accurate estimate of the $\mathcal{T}$-year return
level for every value of $\mathcal{T}$ is given by the distribution $F_Z$ of the parent process, thus making the derivation of the distribution
$F_Y$ of AM superfluous, if the latter requires the preliminary definition of the former.

Results are reported as diagrams of the 5%, 50%, and 95% quantiles of multiplicative bias versus number of years. As
expected, Fig. 4 is in perfect agreement with Figure 7 in Marra et al. (2018), and leads to the same overall conclusions: MEV
exhibits positive bias compared to GEV and GP, but smaller variance. However, Fig. 4 provides an additional result concerning
the performance of the compound parent distribution corresponding to MEV, and shows that both models yield almost identical
results apart from unavoidable sampling fluctuations in the estimation of the 5% and 95% quantiles on 1000 simulated values
of bias $B$. As discussed in Sections 3 and 4, MEV (or more generally $\mathcal{NA}$ distributions) does not add any information with
respect to the parent distribution appearing in MEV formulas. Therefore, once a distribution is selected to describe the ordinary



**Figure 4.** Multiplicative bias for the 100-year return levels obtained from 1000 synthetic samples of varying record length (i.e., number of block/years) and varying number of ordinary events per block/year (10, 50, 100) drawn from Weibull distribution with shape parameters $\kappa = 0.8$ (a-c) and 1.25 (d-f). The reference 100-year return levels are empirically obtained from a $10^5$-year record. Solid lines represent the median bias, while shaded areas (for GEV and MEV) and dashed lines (for GP and Compound parent) represent the 95% Monte Carlo confidence intervals.

events (here, Weibull), its compound version is enough to make statements on any quantile, providing more information than
the derived compound $\mathcal{NA}$ models, which approximate only the upper tail of the (embedded) parent distribution.





## 5.2 Monte Carlo experiment 2: serially dependent processes

This Monte Carlo experiment is designed to study the effect of autocorrelation on $\mathcal{NA}$-based inference. Time series of ordinary events mimicking 3, 5, 10, 20, and 50 years of daily records (i.e., 365 records per year) are simulated $S = 1000$ times to estimate 100-year return levels. The marginal distributions are the same used in the first experiment, i.e., Weibull with shape
parameter $\kappa \in \{0.8, 1.25\}$ and scale parameter $\lambda = 1$. Autocorrelation is modeled by a first-order autoregressive (AR(1)) process with parameter $\rho_1 \in \{0.3, 0.6, 0.9\}$, corresponding to weak, moderate, and relatively high autocorrelation. Weibull-AR(1) time series are generated by CoSMoS framework, which enables the simulation of correlated processes with desired marginal distribution and ACF (Papalexiou, 2018, 2022; Papalexiou and Serinaldi, 2020; Papalexiou et al., 2021, 2023).

Extreme value analysis is performed by GEV for AM, GP for POT of preliminarily declustered data, Weibull-based MEV
for declustered data, Weibull-based $\beta\mathcal{B}C$ for the complete time series. Declustering is based on time lag, selecting the first lag $\tau_0$ such that the empirical ACF becomes smaller than twice the $99\%$ quantile of the sampling distribution of the ACF values under independence. Although this approach is slightly different from that used by Marra et al. (2018), the rationale is the same and it yields $\tau_0$ values that guarantee sufficiently long inter-arrival times as well as a suitable number of events per block for the considered AR(1) ACFs and sample sizes. Sub-sets of ordinary events used for MEV analysis are then defined as
peaks separated by time intervals $\geq \tau_0$. POT for GP analysis are extracted from these sub-sets, while AM for GEV analysis are selected from the original sample, assuming their inter-annual independence. Of course, $\beta\mathcal{B}C$ analysis uses the complete data set and does not require any preliminary declustering procedure as it explicitly accounts for autocorrelation.

Figure 5 compares results of GEV, GP and MEV analysis. For $\rho_1 = 0.3$, values of bias $B$ are similar to those obtained for the previous experiment in Section 5.1 with $\mu_L = 100$ (Fig. 4). This is expected as low values of $\rho_1$ correspond to rapidly
decreasing ACF and therefore $\tau_0 \cong 2 - 3$ time steps, corresponding to sample sizes of ordinary events between about 120 and 180. For $\rho_1 = 0.6$ and 0.9, $\tau_0$ increases to 4-6 and 15-30 time steps, respectively, corresponding to sample sizes of 60-90 and 12-24 ordinary events. The progressively reduced sample size increases MEV uncertainty, which becomes similar to that of GEV and GP models. More importantly, MEV bias dramatically increases with $\rho_1$ and number of years (blocks). The effect of $\rho_1$ is easy to interpret in terms of reduced sample size resulting from declustering with larger $\tau_0$. On the other hand, the effect
of the number of years could appear counter-intuitive as one would expect more accuracy when a larger number of years is available.

Marra et al. (2018) ascribe this behavior "*to uncertain estimation of the weight of the tail of the ordinary events distribution when few data points are used for the fit*". However, this would not be sufficient to explain why the smallest bias corresponds to small numbers of available years, and thus overall smaller samples. The actual issue is the combination of the (average)
number of intra-block peaks (or intra-block sample size; here, $l$ or $\mu_L$), the number of blocks (here, the number of years $n_Y$), and the compounding procedure characterizing MEV.

For fixed $n_Y$, small intra-block sample size $l$ results in great variability of Weibull parameters estimated in each block, which in turn results in heavier tails of compound distributions. As $l$ increases, the inter-block variability of Weibull parameters decreases and the compound distribution resulting from averaging a set of similar Weibull distributions becomes closer and



**Figure 5.** Multiplicative bias for the 100-year return levels obtained from 1000 synthetic samples of varying record length (i.e., number of block/years) from Weibull-AR(1) process with Weibull shape parameters $\kappa = 0.8$ (a-c) and 1.25 (d-f), and AR(1) parameter $\rho \in \{0.3, 0.6, 0.9\}$. The reference 100-year return levels are empirically obtained from a $10^5$-year record. Solid lines represent the median bias, while shaded areas (for GEV and MEV) and dashed lines (for GP and Compound parent) represent the 95% Monte Carlo confidence intervals. MEV and Compound parent distributions are fitted to preliminarily declustered data.





closer to the theoretical Weibull used to simulate. In other words, the compounding mechanism works better in those cases in which it is less required, i.e., when the inter-block variability is small and model averaging (of very similar models fitted on each block) is less justified and useful. On the other hand, when model averaging could be more justified, i.e., when there is substantial uncertainty of the sampling parameters, the spreader is the sampling distribution of parameters the heavier is the tail of the resulting compound distribution, whose shape departs from that of the (true) theoretical distribution.

For given $\mu_L$, when the number of years $n_Y$ is small, compound $\mathcal{N}\mathcal{A}$ models average a small number of components $F_j^{l_j}$, with $j = 1, ..., n_Y$ (e.g., we have three components for $n_Y = 3$ years). In a Monte Carlo experiment, averaging a few heterogeneous components results in a set of heterogeneous compound distributions whose differences tend to compensate on average. Therefore, the Monte Carlo ensembles of compound distributions exhibit high variability and small bias. As $n_Y$ increases, the number of averaging components $F_j^{l_j}$ increases, providing a more accurate picture of the inter-block variability

that is incorporated in the compound distributions. This results in Monte Carlo ensembles of compound distributions with more homogeneous and systematically heavier tails than those of the compound models resulting from small $n_Y$. Therefore, the Monte Carlo ensemble exhibits lower variance and higher bias as $n_Y$ increases for a given $\mu_L$.

As for Fig. 4, Fig. 5 also reports results for the compound distribution of ordinary events, which are almost indistinguishable from those of MEV analysis. Overall, Fig. 5 further confirms the redundancy of MEV models (and more generally, $\mathcal{N}\mathcal{A}$ models)

once we have a compound parent distribution, which has to be estimated in any case to derive $\mathcal{N}\mathcal{A}$ distributions. Moreover, uncorrelated ordinary events resulting from declustering procedures do not guarantee convergence of compound distributions (MEV or parent) to the true distribution. In fact, bias is generally much larger than that of GEV and GP estimates, although the intra-block sample size is generally much larger than that of AM and POT, and the compound distributions have a much larger number of parameters (from 6 to 100, resulting from two-parameter Weibull fitted to one-year blocks over three to 50 years).

Figure 6 compares results of GEV and GP analysis with those of $\beta\mathcal{B}$C and compound parent models. Since $\beta\mathcal{B}$C models (and the corresponding compound parent) use the complete time series instead of declustered data, uncertainty and bias are smaller than those of MEV models (and the corresponding compound parent). Therefore, while time lag declustering seems to yield apparently independent events, the resulting data sets do not provide a faithful description of the upper tail of the true generating process, or better, MEV models do no make a suitable use of these declustered samples. Declustering has negative

effects independently of the intensity of autocorrelation. Of course, larger bias and uncertainty correspond to higher $\rho_1$ values in both MEV and $\beta\mathcal{B}$C analysis. In fact, MEV is affected by significant decrease of sample size due to declustering, while $\beta\mathcal{B}$C suffers from underestimation of ACF, which requires large sample sizes to be reliably estimated (see e.g., Koutsoyiannis and Montanari, 2007; Serinaldi and Kilsby, 2016a). It is worth noting that the GEV and GP results are rather insensitive to autocorrelation. This is expected as the underlying joint dependence structure of AR(1) processes is a Gaussian copula, which

is characterized by asymptotic tail independence and therefore complaint with EVT assumptions. Similarly to Fig. 4 and 5, Fig. 6 shows that the $\beta\mathcal{B}$C model and the corresponding compound parent match (apart from discrepancies due to the issues mentioned above), confirming the redundancy of $\mathcal{N}\mathcal{A}$ models.



**Figure 6.** Multiplicative bias for the 100-year return levels obtained from 1000 synthetic samples of varying record length (i.e., number of block/years) from Weibull-AR(1) process with Weibull shape parameters $\kappa = 0.8$ (a-c) and 1.25 (d-f), and AR(1) parameter $\rho \in \{0.3, 0.6, 0.9\}$. The reference 100-year return levels are empirically obtained from a $10^5$-year record. Solid lines represent the median bias, while shaded areas (for GEV and MEV) and dashed lines (for GP and Compound parent) represent the 95% Monte Carlo confidence intervals. $\beta\mathcal{B}C$ and Compound parent distributions are fitted to complete autocorrelated time series.





### 5.3 Monte Carlo experiment 3: reviewing simulations of Marani and Ignaccolo (2015)

Figure 4 shows that MEV and its compound parent distribution yield a median multiplicative bias $B_{\mathrm{M}} \cong 1.25$ for 100-year
return levels estimated from $n_Y = 50$ years (blocks) of data drawn from Weibull distributions with shape parameter $\kappa = 0.8$ and
average number of events per block $\mu_L \in \{50, 100\}$. On the other hand, $B_{\mathrm{M}} \cong 1.0$ for GEV and GP distributions. For a similar
setup (i.e., $n_Y = 50$, $\kappa = 0.82$, and $\mu_L \in \{30, 100\}$), Marani and Ignaccolo (2015) reported probability plots (probability vs.
quantiles) and relative error

$$R_k = \frac{\hat{x}_k - x_{\mathrm{ref}}}{x_{\mathrm{ref}}}, \tag{9}$$

where $\hat{x}_k$ is the estimate of the target statistics for the $k^{\mathrm{th}}$ Monte Carlo simulation (with $k = 1, ..., S$), and $x_{\mathrm{ref}}$ the reference
(true) value. They found that MEV is almost unbiased, with average relative error $\bar{R} = \frac{\sum R_k}{S} \cong 0$, while GEV exhibits bias,
with $\bar{R} \cong 5\%$ and $\cong 30\%$ for the 100-year and 1000-year return levels, respectively. On the other hand, for the 100-year return
level, simulations in Section 5.1 (reproducing those of Marra et al. (2018)) yield $\bar{R} \cong 25\%$ for MEV and $\bar{R} \cong 0$ for GEV.
Therefore, we re-run Monte Carlo simulations described by Marani and Ignaccolo (2015) to understand the reason of such a
disagreement. We anticipate that the foregoing discrepancies depend on the misuse of methods used to summarize multi-model
ensembles. Thus, before describing Monte Carlo experiments and their outcome, we need to recall some theoretical concepts
that are required to correctly interpret numerical results.

### 5.3.1 Summarizing multi-model ensembles: some overlooked concepts

Monte Carlo simulations are usually used to study the uncertainty affecting estimates based on finite-size samples (that provide
incomplete information about the underlying process) or to approximate population distributions (or statistics) when mathemat-
ical closed-form expressions are not available. Examples of these applications are the experiments reported in Sections 5.1 and
5.2, and the Markov Chain Monte Carlo (MCMC) simulations performed in Bayesian inference to obtain posterior distributions
of model parameters with unknown mathematical form.

In all cases, the primary output of Monte Carlo simulations is a set of parameters identifying a set of models (multi-model
ensemble) that is then used to estimate the target statistics of interest. For example, simulations of $S$ finite-size samples in
Sections 5.1 and 5.2 are used to fit a set of $S$ GEV distributions. These are then used to calculate a set of $S$ 100-year return
levels, which are eventually used to build confidence intervals.

However, a multi-model ensemble can be summarized in many different fashions. This aspect is quite known in Bayesian
inference where MCMC posterior distribution of model parameters yields a set of models, and these models need to be sum-
marized in some way to obtain a representative point estimate of a statistic of interest (e.g., Renard et al., 2013; Fawcett and
Walshaw, 2016; Fawcett and Green, 2018). Let $S$ be the number of Monte Carlo replications, $F(z|\boldsymbol{\theta}_k)$ (with $k = 1, ..., S$) the
$k^{\mathrm{th}}$ member of the Monte Carlo multi-model ensemble (e.g., the $k^{\mathrm{th}}$ Weibull distribution fitted to the $k^{\mathrm{th}}$ simulated sample), $z_p$
a target quantile with nonexceedance probability $p$, and let define the quantile function as the inverse of the cdf, $Q = F^{-1}$. A
representative point estimate of $z_p$ can be for instance the mode of the $S$ quantiles $z_{p,k} = Q(p|\boldsymbol{\theta}_k) = F^{-1}(p|\boldsymbol{\theta}_k)$.





More popular point estimates of $z_p$ (or whatever statistics) rely on the definition of so-called predictive distributions and predictive quantile functions. The sampling predictive cdf reads as

$$
\begin{aligned}
\bar{F}(z) :=& \frac{1}{S} \sum_{k=1}^{S} F(z|\boldsymbol{\theta}_k) \\
\cong& \mathbb{E}_{\Omega_{\boldsymbol{\theta}}}[F(z|\boldsymbol{\theta})],
\end{aligned}
\tag{10}
$$

and the corresponding quantile with specified nonexceedance probability $p$ is given by

$$
\begin{aligned}
z_{p,\bar{F}} =& \left\{ z : \frac{1}{S} \sum_{k=1}^{S} F(z|\boldsymbol{\theta}_k) = \bar{F}(z) = p \right\} \\
=& \bar{F}^{-1}(p).
\end{aligned}
\tag{11}
$$

The sampling predictive quantile function reads as

$$
\begin{aligned}
\bar{Q}(p) :=& \frac{1}{S} \sum_{k=1}^{S} Q(p|\boldsymbol{\theta}_k) \\
\cong& \mathbb{E}_{\Omega_{\boldsymbol{\theta}}}[Q(p|\boldsymbol{\theta})] = \mathbb{E}_{\Omega_{\boldsymbol{\theta}}}[F^{-1}(p|\boldsymbol{\theta})],
\end{aligned}
\tag{12}
$$

resulting in predictive quantile estimates

$$
\begin{aligned}
z_{p,\bar{Q}} =& \frac{1}{S} \sum_{k=1}^{S} F^{-1}(p|\boldsymbol{\theta}_k) \\
=& \overline{F^{-1}(p)}.
\end{aligned}
\tag{13}
$$

Let us denote the empirical cdf and quantile function of the $S$ sampled quantiles $z_{p,k}$ as $F_S$ and $Q_S$, respectively. Recalling that the distribution of $z_p$ can be approximated by the distribution of order statistics, and the latter is a beta distribution (see Eq. 1), we can write $F_S(z_p) \cong F_\beta(F(z)|pS', (1-p)S')$, where $S' = S+1$. Therefore, the foregoing $z_p$ estimators can be complemented by the median estimator defined as

$$
\begin{aligned}
z_{p,\mathrm{M}} =& \{ z : \mathbb{P}[Z_{p,k} \leq z] = F_S(z_p) = 0.5 \} \\
=& F_S^{-1}(0.5) \cong F^{-1}\left( F_\beta^{-1}(0.5|pS', (1-p)S') \right).
\end{aligned}
\tag{14}
$$

Similarly, we can also define the median probability of a fixed quantile $z_p$ from an ensemble of cdfs as follows

$$
\begin{aligned}
p_{\mathrm{M}} =& \{ p : Q_S(0.5) = z_p \} \\
=& \{ p : F_S(z_p) = 0.5 \} \\
\cong& \{ p : F_\beta(F(z_p)|pS', (1-p)S') = 0.5 \}
\end{aligned}
\tag{15}
$$

The foregoing formulas indicate that the three $z_p$ estimators obviously represent different quantities. Focusing on $z_{p,\bar{F}}$ and $z_{p,\bar{Q}}$, and comparing Eqs. 11 and 13 we have that

$$
\bar{F}^{-1}(p) \neq \overline{F^{-1}(p)}.
\tag{16}
$$





Eq. 16 is the sampling counterpart of $Q(\mathbb{E}[F(z_p)]) \neq \mathbb{E}[Q(F(z_p))] \equiv \mathbb{E}[Z_p]$, which in turn follows from the well-known general inequality

$$\mathbb{E}[F(z)] \neq F(\mathbb{E}[Z]), \tag{17}$$

stating that the distribution of the expected value of $Z$ is different from the expected distribution of $Z$. In fact, since $F$ is commonly a nonlinear transformation of $Z$ (as well as of the parameters $\boldsymbol{\theta}$), it hinders the interchangeability of the (linear) expectation operator $\mathbb{E}$. In passing, such an inequality also partly caused a long 'querelle' on plotting position formulas (see e.g., Makkonen, 2008; Cook, 2012; Makkonen et al., 2013).

On the other hand, $z_{p,\mathrm{M}}$ is the only estimator that guarantees the identity between the $z_p$ estimates obtained from ensembles of $Q$ or $F$ functions. This property depends on the fact that the median (as well as every quantile) is a rank-based (central tendency) index, and ranking is a transformation that does not depend on absolute values, and therefore passes unaffected trough nonlinear monotonic functions such as $Q$ and $F$. This means that the median parameters $\boldsymbol{\theta}_\mathrm{M}$ correspond to $z_{p,\mathrm{M}}$ and $p_\mathrm{M}$. This property does not hold for the expectation operator $\mathbb{E}$. In fact, generally $F(z_p|\mathbb{E}[\boldsymbol{\theta}]) \neq \mathbb{E}[F(z_p|\boldsymbol{\theta})]$.

The foregoing concepts and properties play a key role for the correct interpretation of results reported in the next section.

### 5.3.2 Numerical simulations: the consequences of overlooking theory

Marani and Ignaccolo (2015) supported the introduction of MEV by five Monte Carlo experiments (referred to as cases 'A', 'B', 'C', 'A2', and 'B2'), comparing the accuracy of MEV to that of standard asymptotic models of BM (i.e., Gumbel and GEV distributions). For the cases 'B' and 'B2', Marani and Ignaccolo (2015) did not provide enough information to enable their replication. Therefore, we focused on cases 'A', 'C', and 'A2', which are sufficient to support our discussion. Case 'A' consists of simulating $S = 1000$ samples from a Weibull distribution with scale parameter equal to 7.3, shape parameter $\kappa = 0.82$, number of blocks (years) $n_Y = 50$, and number of events per block (here, wet days per year) $l = 100$. Case 'C' is similar to 'A', the only difference being that the number of events per block is drawn from a uniform distribution $\mathcal{U}(21, 50)$. The setup of case 'A2' is similar to that of 'A'; however, it explores the effect of varying $l$ from 10 to 200 by steps of 10 events per block. Therefore, Gumbel, GEV, and MEV distributions of BM are fitted to each of the $S$ samples. For the cases 'A' and 'C', the accuracy of the three models is assessed by comparing "*the ensemble average distributions,* $\zeta_{\mathrm{MEV}}(y)$, $\zeta_{\mathrm{GEV}}(y)$, $\zeta_{\mathrm{GUM}}(y)$ *as the means of the distributions of $Y$ computed over the 1000 synthetic time series*" (Marani and Ignaccolo, 2015). For the case 'A2', the three models are evaluated in terms of average relative error $\bar{R}$ of the estimates of the 100- and 1000-year return levels. The reference (true) return levels are empirically obtained from $10^6$ years of simulated samples.

Figures 7a-c reproduce Figures 3a,c in Marani and Ignaccolo (2015). For the case 'A', we used both $\kappa = 0.82$ and 0.73 as the original parametrization cannot reproduce results of Figure 3a in Marani and Ignaccolo (2015). In fact, analyzing the original Figure 3a, the reference 100- and 1000-year return levels should be close to 150 and 210, respectively, while $\kappa = 0.82$ yields values close to 109 e 143, which in turn are consistent with case 'C'. Therefore, we used $\kappa = 0.73$ to obtain a figure as close as possible to the original one. Nonetheless, the exact value of $\kappa$ is inconsequential in the following discussion, and we use both $\kappa = 0.82$ and 0.73 for completeness.

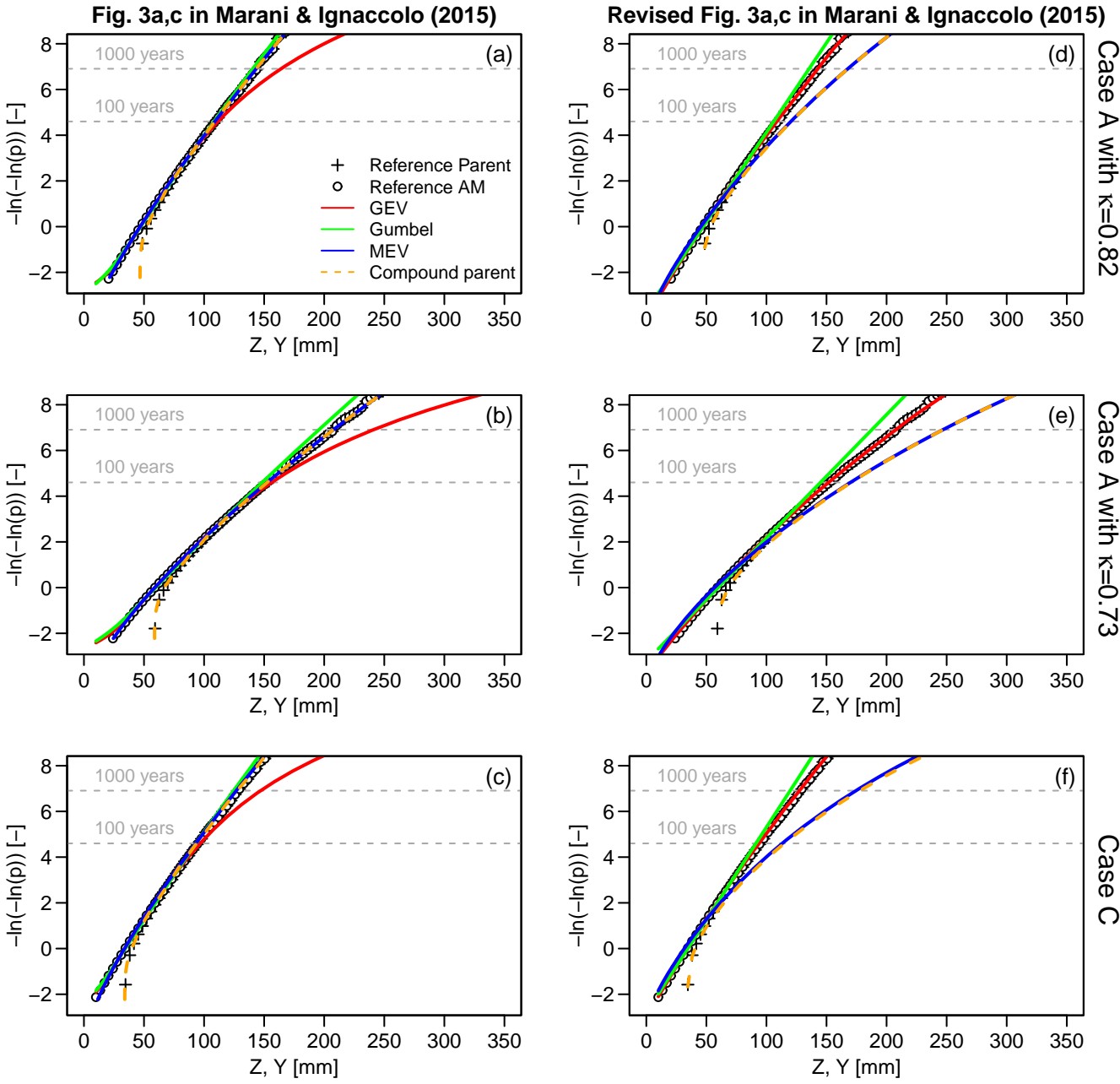

**Figure 7.** Probability plots (probability vs. quantile) showing different models for AM $Y$ resulting from the Monte Carlo experiments denoted as cases 'A' (a,b,d,e) and 'C' (c,f) (see main text for details about the simulation setup). Panels (a-c) reproduce results reported in Marani and Ignaccolo (2015, Figures 3a,c), while panels (d-f) show the revised version with corrections accounting for inconsistencies in the calculation of compound quantiles and misuse of multi-model ensemble averaging.





The key aspects in Fig. 7a-c are (i) the perfect match of MEV and its compound parent, confirming the redundancy of $\mathcal{NA}$ models when their parents are already known, and (ii) the accuracy of MEV and its compound parent against the prominent bias of GEV, which contrasts results reported by Marra et al. (2018) and in the previous sections. The reason of such a discrep-

ancy is that Fig. 7a-c (and Figure 3a in Marani and Ignaccolo (2015)) do not show what they are supposed to do, thus making the comparison unfair and misleading. In fact, contrary to the description in Marani and Ignaccolo (2015), the MEV curves in Fig. 7a-c do not refer to the predictive MEV obtained by averaging $S$ MEV distributions according to Eqs. 10 and 11. Instead, recalling that MEV is itself a predictive distribution (i.e., the average of multiple components $F_j^{l_j}$, with $j = 1, ..., n_Y$; see Section 2), MEV curves in Fig. 7a-c refer to the predictive quantile functions (over $S$ samples) of the predictive quantile functions

(over $n_Y$ samples) associated to MEV structure. In other words, Fig. 7a-c report the pairs $(z_{p,\bar{Q}}, p)$ instead of the claimed $(z_{p,\bar{F}}, \bar{F})$, and these pairs differ from each other (see Section 5.3.1). In more detail, $z_{p,\bar{Q}} \cong \mathbb{E}_S[\mathbb{E}_{\Omega_{\boldsymbol{\theta}_S}}[(F_{\text{WEI}}^l)^{-1}(p|\boldsymbol{\theta}_S)]]$, while the figure should show $z_{p,\bar{F}}$ obtained by inverting $\bar{F} \cong \mathbb{E}_S[F_{\text{MEV}}] = \mathbb{E}_S[\mathbb{E}_{\Omega_{\boldsymbol{\theta}_S}}[(F_{\text{WEI}}^l)(z_p|\boldsymbol{\theta}_S)]]$.

On the other hand, Fig. 7a-c (and Figures 3a,c in Marani and Ignaccolo (2015)) correctly show the predictive distributions of Gumbel and GEV. However, this hinders a fair comparison. In fact, EVT states that the asymptotic model of BM is a GEV

distribution (under suitable conditions) and not the compound version of GEV resulting from averaging $S$ GEV models. Such a compound GEV distribution has always a larger variance and heavier tails than its classical GEV counterpart (see discussion in Section 6). Therefore, to be consistent with EVT, the ensemble of GEV and Gumbel distributions should be summarized using a transformation, such as the median, that retains the expected GEV/Gumbel shape. Figures 7d-f show the median GEV and Gumbel distributions along with the actual predictive MEV (as it should be). Results in Fig. 7d-f are fully consistent with

those reported by Marra et al. (2018) and in Sections 5.1 and 5.2, confirming the low bias of asymptotic models and the natural tendency of compound distributions to exhibit heavier tails than their components and their generating processes. Moreover, the perfect agreement of the upper tail of MEV and that of compound parent distributions in Fig. 7d-f further confirms (if still needed after many examples) the redundancy of $\mathcal{NA}$ models once their parent distributions are defined, which means that such models are useless in practical applications.

Similar remarks hold for the case 'A2'. Results in Fig. 8a,b are close to those reported by Marani and Ignaccolo (2015) in their Figure 4a, with MEV showing $\bar{R} \cong 0$ for both 100- and 1000-year quantiles, and GEV showing $\bar{R} \cong 0$ for 100-year return level and $\bar{R} \cong 5\%$ for 1000-year return level. Gumbel distribution yields slightly negative $\bar{R}$ for both return levels with smaller values for higher $\kappa$, which corresponds to a generating Weibull distribution closer to exponential, thus allowing faster convergence to the first asymptotic distribution of EVT. As for the cases 'A' and 'C', these results are affected by mixing

predictive distributions and predictive quantile functions as well as the improper use of the former to summarize the ensemble of GEV and Gumbel models. Figures 8c,d show $\bar{R}$ values corresponding to true predictive MEV and median GEV and Gumbel distributions preserving distribution shape. As expected, the GEV model correctly describes BM, while the compound structure of MEV yields heavier tails. Once again, results from MEV and compound parent are almost indistinguishable due to the redundancy of MEV (and any $\mathcal{NA}$ model in general).

The work by Marani and Ignaccolo (2015) also suffers from several mismatches between text and figures. For example, concerning the case 'A2' and the corresponding Figure 4a, they state "*GEV approach systematically overestimates the 100-yr*



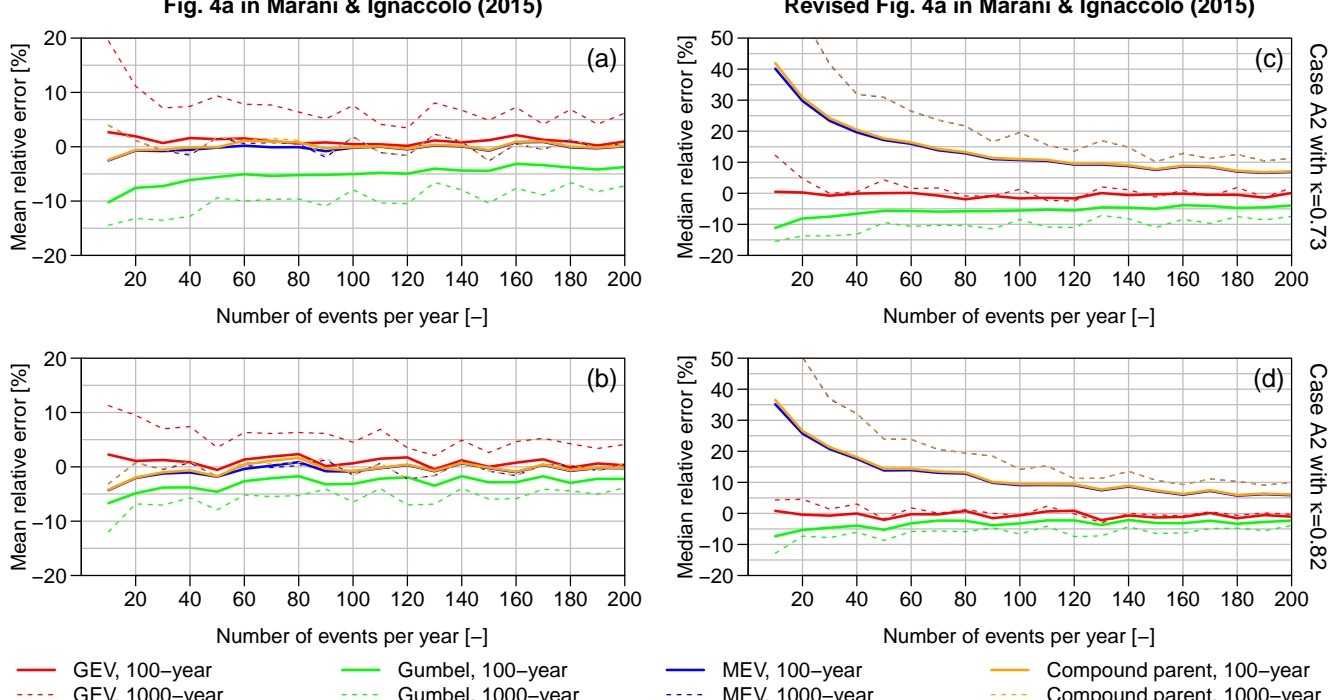

**Figure 8.** Relative errors for 100- and 1000-year return levels resulting from the Monte Carlo experiment denoted as case 'A2' (see main text for details about the simulation setup). Panels (a,b) reproduce results reported in Marani and Ignaccolo (2015, Figure 4a) for $\kappa \in \{0.73, 0.82\}$, while panels (c,d) show the revised version with corrections accounting for inconsistencies in the calculation of compound quantiles and misuse of multi-model ensemble averaging.

*extreme rainfall intensity by 5% even for large numbers of wet days. The Gumbel approach systematically underestimates the 100-yr extreme rainfall intensity by about 5%. For the 1000-years return period intensities, the GEV approach severely overestimates actual extreme events (minimum relative error is 30% for n = 200 events/year) whereas the Gumbel approach*

*yields underestimation errors of about 10%*". However, in contrast with the text, their Figure 4a shows that GEV has $\bar{R} \cong 0$ for 100-year return level, and $\bar{R} \cong 10\%$ for the 1000-year return level, while Gumbel distributions have $\bar{R} \cong -15\%$ and $\cong -30\%$ for the 100- and 1000-year return levels, respectively. Concerning the case 'B2' and the corresponding Figure 4b, any interpretation is impossible as Figure 4b in Marani and Ignaccolo (2015) reports "*Root Mean Square % Error*" whereas the text refers to $\bar{R}$, and it is not even clear if Figure 4b actually refers to the case 'B2'.





## 6 Discussion

The proposal of $\mathcal{NA}$ models as an alternative to classic EVT models suffers from some problems that seem to be quite widespread in the hydrological literature dealing with statistical methods (see e.g., discussions in Serinaldi and Kilsby, 2015; Serinaldi et al., 2018, 2020a, 2022b):

1. Data analysis should be supported by preliminary scrutiny of its rationale, allowing for instance the recognition of the 'circular reasoning' affecting practical use of $\mathcal{NA}$ models of BM. Extreme value models are powerful tools if applied in the right context according to their motivation and assumptions. Their usefulness relies on the fact that they provide an approximate description of the upper (or lower) tails of the distribution of parent processes when the latter is unknown and there are no data (or data are not enough) to reliably estimate it. $\mathcal{NA}$ models of BM contradict this principle. In fact, $\mathcal{NA}$ models require the preliminary estimation of a parent distribution $F_Z$ to build a surrogate distribution $F_Y$ that approximates a tail of $F_Z$, neglecting that $F_Z$ is already known/fitted.

   For example, Marra et al. (2023) studied the distribution of worldwide daily rainfall data over low/moderate thresholds showing that a Weibull model provides a good fit and reproduces L-moments of AM even when AM are excluded from calibration. Conversely, using GP tails provides the same results only over the 95% threshold and overestimates the heaviness of the upper tail when the GP model is assumed for low/moderate thresholds (in agreement with results reported by Serinaldi and Kilsby (2014b) about Multiple Threshold Method (Deidda, 2010)). The natural interpretation of these results would be that the Weibull distribution is a good model $F_Z$ for the parent process $Z$ (positive rainfall or rainfall over low/moderate thresholds) confirming previous results reported in the literature, while GP model works well for exceedances over high thresholds (as postulated by EVT), and does not work well (as expected) for low/moderate thresholds, that is, outside its range of validity. Recalling the theoretical link between GP and GEV, this also means that the latter is a good model for rainfall BM.

   For practical applications, this should translate into the following recommendations: (i) use GEV if only BM are available (e.g., AM from hydrologic reports), and (ii) use $F_Z$ (e.g., (compound) Weibull) if you have information on $Z$, which can be either the process of all positive rainfall or rainfall over arbitrary low/moderate thresholds if the latter is deemed easier to fit. In the latter case, calculate the $\mathcal{T}$-year return levels as the $(1 - \frac{\mu}{\mathcal{T}}) \cdot 100\%$ quantiles of $F_Z$, where $\mu$ is the (mean) inter-arrival time (in years) between two observations of $Z$ (e.g., Serinaldi, 2015; Volpi et al., 2019).

   Such a plain reasoning highlights that there is no need to build an additional distribution of BM (i.e., (compound) $F_Z^l$), in the same way we do not need to define the GEV distribution of AM once we already inferred a GP model of POT. Nonetheless, Marra et al. (2023) interpreted their results as evidence to support $\mathcal{NA}$ models of BM, missing that the fitted Weibull distributions over zero, low or moderate thresholds are conceptually similar to each other and can be used directly to make inference about any desired quantile without deriving redundant models of BM (here, exponentiated Weibull).





2. New methods need to be suitably validated before being applied. Actually, applications to real-world data are often improperly used as validation. Proper validation/falsification requires the use of processes with known properties that match or contrast the model assumptions. For example, $\mathcal{NA}$ models, such as (S)MEV, have only been assessed for parent processes with known marginal distributions under independence (e.g., Marra et al., 2018), while the effect of dependence and the effectiveness of declustering were not checked. We encourage modelers to perform proper Monte Carlo simulations, as suitable methods are readily available for such a kind of analysis (e.g., Serinaldi and Lombardo, 2017a, b; Papalexiou, 2018; Serinaldi and Kilsby, 2018; Koutsoyiannis, 2020; Papalexiou and Serinaldi, 2020; Papalexiou et al., 2021; Papalexiou, 2022, among others). Of course, numerical experiments should be supported by the necessary theoretical knowledge allowing correct implementation and interpretation, and preventing inconsistencies such as those discussed for instance in Section 5.3.

On the other hand, proper validation was replaced by quite an extensive use of cross-validation exercises on observed data (e.g., Miniussi and Marani, 2020; Mushtaq et al., 2022), which might however be misleading because:

(a) Hydro-climatic records come from processes with inherently unknown properties as only estimates of the variables of interest are available.

(b) Cross-validation is usually performed on short time series (commonly, a few years of data), and model estimates (from shorter calibration sub-sets) are compared with sample estimates (from shorter verification sub-sets), which might be not representative of the true value of the target statistics. Cross-validation relies on the assumptions that the calibration sub-sets are representative of the population, and out-of-sample sub-sets come from the same population. However, for autocorrelated processes, very long time series might be required to explore the state space of the studied process, thus meaning that the observed series might be not representative, especially when focusing on extreme values. In hydro-climatic processes, this issue is exacerbated by the effect of long term fluctuations characterizing the climate system at local and global spatial scales.

(c) Standard bootstrap resampling used in cross-validation might also be misleading. In fact, it provides correct results under the assumption that the state space is explored under independence and therefore relatively short samples are enough to give reliable picture of the range of possible outcomes. If the hypothesis of independence is not valid, the observed values might cover a sub-set of the state space, and the standard bootstrap commonly applied in MEV literature just conceals this fact.

3. Often, inappropriate validation and iterated application to real-world data generate quite an extensive literature confusing numerical artifacts with physical properties (see e.g., Serinaldi and Kilsby, 2016a; Serinaldi et al., 2020a, 2022b, for paradigmatic examples). Such a literature is often improperly used to support a given method by arguments like 'there is such a strong scientific body of literature demonstrating the technical advantages of these approaches'. However, consensus is not a scientific argument. Historically, the main scientific progresses occurred when some one called into question widely accepted mainstream theories using arguments more solid than those of the superseded theories. Consensus is even more questionable when a method is iteratively applied without a necessary neutral/independent validation. The





literature on $\mathcal{NA}$ models tends to suffer from these problems, and our discussion in Section 5.3 illustrates how these models have been iteratively applied without the above-mentioned independent analysis. It is quite common reading sentences such as 'these new approaches have been shown to be practically useful under real conditions, that are showing their practical advantage over traditional methods'. Such a kind of statements do not provide any technical information about either the relationship between the distribution of BM and POT and their corresponding parent or the rationale and effects of compounding multiple models, or the difference between the parametrization of GEV and $\mathcal{NA}$ models, for instance. Moreover, if a method is biased, as shown in the previous sections, multiple applications to real-world data do not make it unbiased.

4. Often, (seemingly) new methods are not put in their broader context, and are denoted by uninformative names, thus concealing their nature and hindering correct interpretation. In particular, $\mathcal{NA}$ distributions are just special versions of the class of compound distributions (e.g., Dubey, 1970; van Montfort and van Putten, 2002)

$$
\begin{aligned}
\tilde{f}(x) &= \int_{\Omega_{\boldsymbol{\theta}}} f(x, \boldsymbol{\theta}) \mathrm{d}\boldsymbol{\theta} \\
&= \int_{\Omega_{\boldsymbol{\theta}}} f(x|\boldsymbol{\theta}) f(\boldsymbol{\theta}) \mathrm{d}\boldsymbol{\theta} \\
&= \mathbb{E}_{\Omega_{\boldsymbol{\theta}}}[f(x|\boldsymbol{\theta})],
\end{aligned}
\tag{18}
$$

where $\tilde{f}(x)$ is the marginal pdf of a generic variable $X$, $f(\boldsymbol{\theta})$ is the pdf of the parameter vector $\boldsymbol{\theta}$ of the distribution $f(x|\boldsymbol{\theta})$, and $\Omega_{\boldsymbol{\theta}}$ is the state space of $\boldsymbol{\theta}$ when it is treated as a random variable $\boldsymbol{\Theta}$. The variance $\mathbb{V}[X]$ of $\tilde{f}(x)$ is always greater than that of its components $f(x|\boldsymbol{\theta})$, as it is (e.g., Karlis and Xekalaki, 2005)

$$
\mathbb{V}[X] = \mathbb{E}_{\Omega_{\boldsymbol{\theta}}}[\mathbb{V}_{X|\boldsymbol{\theta}}[X]] + \mathbb{V}_{\Omega_{\boldsymbol{\theta}}}[\mathbb{E}_{X|\boldsymbol{\theta}}[X]].
\tag{19}
$$

Compound distributions have been presented in the literature under various names and contexts, such as 'superstatistics' in physics and hydrology (Beck, 2001; Porporato et al., 2006; De Michele and Avanzi, 2018), 'predictive distributions' in theoretical and applied statistics (Benjamin and Cornell, 1970; Wood and Rodríguez-Iturbe, 1975; Stedinger, 1983; Bernardo and Smith, 1994; Kuczera, 1999; Coles, 2001; Cox et al., 2002; Gelman et al., 2004; Renard et al., 2013; Fawcett and Walshaw, 2016; Fawcett and Green, 2018), or without introducing any specific name (Koutsoyiannis, 2004; Allamano et al., 2011; Botto et al., 2014; Yadav et al., 2021). In more detail, Eq. 18 *"might be referred to as the prior (Bayesian) distribution or the posterior (Bayesian) distribution on $X$, depending on whether a prior or posterior distribution of $\theta$ is used to determine $\tilde{f}(x)$"* (Benjamin and Cornell, 1970, pp. 632-633). $f(\boldsymbol{\theta})$ can be analytical (e.g., Skellam, 1948; Moran, 1968; Dubey, 1970; Hisakado et al., 2006), or empirical, resulting from Monte Carlo simulations, bootstrap resampling, or estimation from multiple sub-samples, such as in the case of $\beta\mathcal{B}C$ or MEV inference.

However, using our notation, $\tilde{f}(x)$ *"can be interpreted as a weighted average of all possible distributions $f(x|\boldsymbol{\theta})$ which are associated with different values of $\boldsymbol{\theta}$. In this sense* [Equation 18] *can be interpreted as an application of the total*





*probability theorem... In any event we note that the unknown parameter will not appear in $\tilde{f}(x)$, as it has been "integrated out" of the equation. We also note that as more and more data become available, the distribution of $\boldsymbol{\theta}$ will be becoming more and more concentrated about the true value of the parameter. We should generally expect the distribution $\tilde{f}(x)$ to* 655 *be wider, e.g., to have a larger variance, than the true $f(x)$, since the former incorporates both inherent and statistical uncertainty*" (Benjamin and Cornell, 1970, pp. 632-633).

In other words, $\mathcal{NA}$ models, such as $\beta\mathcal{B}$C and MEV, are just the output of what is often referred to as multi-model ensemble averaging (e.g., Burnham and Anderson, 2002; Giorgi and Mearns, 2002, and references therein). The inherent nature of compounding/averaging procedures explains the tendency of $\mathcal{NA}$ models to yield $\tilde{f}(x)$ with tails heavier than 660 those of the true underlying distribution $f(x)$, and progressive convergence of $\tilde{f}(x)$ to $f(x)$ as the (block) sample size increases and $f(\boldsymbol{\theta})$ becomes more and more concentrated around the true value of the parameter(s). It also clarifies that the properties of $\mathcal{NA}$ models of BM depend on being compound models rather than extreme value models. In fact, same results can be obtained by directly compounding the distributions of the parent process without any additional derivation of the corresponding distributions of BM. Furthermore, recognizing the rationale of compound models allows us to 665 understand that the BM process is different from the parent one, and the distribution of the former is useful only if latter is not available. Finally, as shown in Section 5.3, understanding the nature of compounding procedures is fundamental to correctly summarize and interpret multi-model outputs.

## 7 Conclusions

This study presented an inquiry on non-asymptotic ($\mathcal{NA}$) distributions $F_Y$ of block maxima (BM) $Y$, which was motivated by 670 their increasing use in data analysis without a necessary preliminary validation/falsification under controlled conditions, and a deep discussion of their rationale and relationship with the distribution $F_Z$ of the generating process $Z$. We discussed their redundancy and practical uselessness in real-world analysis. This apparently bold statement relies on very basic facts: (i) the distribution $F_Z$ of a process $Z$ provides all information about any quantile or summary statistics (extreme or not); (ii) extreme value distributions $F_Y$ of BM corresponding to the parent process $Z$ are just approximations of the tails of the distribution $F_Z$, 675 and they have a role only if $F_Z$ is unknown; and (iii) $\mathcal{NA}$ distributions require the preliminary knowledge/estimation of $F_Z$; however, once $F_Z$ is known or fitted to data, $\mathcal{NA}$ distributions of BM are no longer needed, and their derivation is superfluous as $F_Z$ already provides all information. In this context, the use of asymptotic extreme value models is justified by the fact that they do not require the preliminary knowledge or estimate of $F_Z$ (under suitable conditions).

While the foregoing logical arguments should be sufficient to call into question the practical use and usefulness of $\mathcal{NA}$ 680 models, we further demonstrated these issues by simplified examples, re-analysis of real-world data, and suitable Monte Carlo simulations. The aim was to support conceptual statements with numerical experiments that are easy to reproduce and can independently be checked. In this way, debate can be based on technical counter-arguments and proper analysis of data drawn from processes with known properties, avoiding the 'consensus' argument, and resetting the discussion about $\mathcal{NA}$ models within the boundaries of the scientific method.

Of course, the questionable usefulness of $\mathcal{NA}$ models in practical applications does not mean that they are not useful at all. As shown in this study and by Serinaldi et al. (2020b), $\mathcal{NA}$ formulation clarifies the inherent relationship between the distribution of BM ($F_Y$) and that of their generating process ($F_Z$), thus shedding light on some inferential aspects from a theoretical point of view. For example, $\mathcal{NA}$ formulation highlights that the difference between return periods/levels estimated from $F_Y$ and $F_Z$ does not depend on sample size (as incorrectly stated in the literature) but on the theoretical difference

of the processes $Y$ and $Z$, and cannot be reduced. $\mathcal{NA}$ expressions also allow a better understanding of the mechanism of compounding distributions to account for multiple generating processes, showing the dualism of additive and multiplicative mixing in the derivation of $F_Y$ from $F_Z$ (Serinaldi et al., 2020b). In principle, $\mathcal{NA}$ models incorporating dependence are also the basis for the theoretical study of the corresponding asymptotic models free from preliminary definition of $F_Z$.

To conclude, models and methods should be thought and used in the right context and for suitable purposes. Reliability

of models must rely on a careful preliminary analysis of their consistency with logic, theory, data, processes analyzed, and problem at hand. A cautious approach should start from the assumption that a new model is likely questionable in terms of novelty (it can already exist, perhaps under a different name in different disciplines), theoretical correctness, and practical usefulness. Therefore, model developers should perform a deep literature review (possibly extended to other disciplines), clearly understand rationale, assumptions, and purpose of the model, and attempt model falsification rather than validation.

New models should be tested under controlled challenging conditions. We believe that these recommendations are cornerstones of a rigorous scientific inquiry and are too often neglected. Calling into question the practical usefulness of $\mathcal{NA}$ models of BM is precisely an application of that investigation method.

*Data availability.*  Data are freely available from University of Hawaii Sea Level Center (UHSLC) repository
(Caldwell et al., 2015, http://uhslc.soest.hawaii.edu/data/?rquh745a/).

*Author contributions.*  FS: Conceptualization, Methodology, Software, Formal analysis, Writing - original draft, Writing - review & editing,
Visualization. FL: Conceptualization, Methodology, Writing - review & editing. CGK: Conceptualization, Writing - review & editing.

*Competing interests.*  The authors declare that they have no known competing financial interests or personal relationships that could have appeared to influence the work reported in this paper.

*Acknowledgements.*  Francesco Serinaldi and Chris G. Kilsby acknowledge the support from the Willis Research Network. Federico Lombardo is grateful to the Italian National Fire and Rescue Service for the continuous support. The authors thank Dr. Theano Iliopoulou
(National Technical University of Athens), Dr. Marco Marani (Università degli Studi di Padova), and Dr. Giuseppe Mascaro (Arizona State University) for their critical remarks that helped substantially improve content and presentation of a previous version of the manuscript. The analyses were performed in R (R Development Core Team, 2022).





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
