# Peer review of "Non-asymptotic distributions of water extremes: much ado about what?"

_Hydrology and Earth System Sciences, 2023_

## Referee Comment (RC2)

Review of hess-2023-234

This manuscript critics the use of non-asymptotic (NA) distributions of block maxima. The authors bring two main arguments to support their critic. The first is that NA require knowledge of the parent distribution and that, when this is known, there would be no need for deriving distributions of block maxima. The second is that the presence of serial correlation in the observations would decrease the potential advantage of NA. The manuscript then follows with some targeted comments to specific studies.

The paper addresses a relevant topic, but the manuscript falls short at supporting its main conclusions. This is not due to technical errors, rather to a narrow (mono-disciplinary) vision of the problems at hand (see main comments) and to an erroneous generalization of case-specific objections. Since all statistical models present advantages and disadvantages, which depend on how much the underlying assumptions are met/not met (or, to quote the statistician George Box: "all models are wrong, but some are useful"), I believe that the presented critics cannot be generalized to NA methods as a whole. Rather, they should be targeted to highlight specific aspects of NA methods that need attention and/or the specific NA approaches that need attention.

Considering the main comments below, I believe the manuscript should be deeply revised before reconsideration. I am not sure this can be handled within a major revision, because the take-home messages would need important adjustments. The title should be revised to be more pertinent with the actual outcomes. This also pertains the title of some sections/subsections which border disrespectfulness (e.g., section 3, 4, 5). The manuscript is long and contains numerous repetitions. It includes unclear and/or incorrect reasoning in some sections, which prevent from fully understanding some parts (see comments 6 and 7 below). It could be halved in length without changing the message.

All the references in this document can be found in the preprint of the manuscript.

Given my lack of specific expertise, comment 6 was written with the help of a colleague expert in Bayesian statistics.

I hope my comments are helpful.
Kind regards,
Francesco Marra

Main comments

1. My main comment turns out to be a citation from the manuscript itself (line 694):
   "*models and methods should be thought and used in the right context and for suitable*

*purposes*". This sentence in the conclusions contradicts several of the arguments presented in the manuscript. The authors proceed for 30 pages (pages 1-30) repeatedly claiming that NA methods are 'superfluous'. Only in lines 685-694 they contradict this argument stating that what they report in the manuscript "*does not mean that they are not useful at all*". They then proceed listing what, in their view, are potentially useful applications of NA approaches and finish off with the citation I started with. I may add that the reported list of 'useful' applications is limited by the imagination of the authors (as shown by the sole presence of self-references in here) and, mostly, by the perspective they adopt. In this, it seems they forget the gap that there exists between theory and practice, between advancements in theory and practical use of extreme value distributions of any kind by hydrologists, risk modelers and end users in general. In fact, several applications of NA methods follow the directions accepted as 'useful' by the authors, and other applications of NA methods follow directions that are useful, although not within the directions imagined by the authors. One notable example is the connection between physical processes and statistics, which can only exist in a NA model, given that real world physics is not asymptotic. It is my believe that physical processes should direct the statistics we use. The physics of the processes we are dealing with is not asymptotic. At this concern I must cite again the authors (lines 623-624): "*Historically, the main scientific progresses occurred when some one called into question widely accepted mainstream theories using arguments more solid than those of the superseded theories*". It almost seems we think alike on this point, although with different concepts for 'mainstream'.

2. One argument is that NA methods require the knowledge of the parent distribution and that when this is known, there would be no need for deriving a block maxima (BM) distribution. This is technically true but seems to neglect situations in which a BM distribution is helpful (even though, I agree, not technically essential). Some examples: empirical comparison with observations of BM only; fair comparison with estimates from EVT distributions; providing information that practitioners can use without changing habits. Doing these directly from the compound parent, although possible, would be troublesome and possibly confusing for non-experts.

3. The issue with serial correlation is important and could affect some applications of NA methods. I believe future NA applications (either for block maxima or directly from the parent distribution) should keep this in mind. In this, the paper is a relevant addition to the literature. Still, it falls short at supporting the adjective 'superfluous' that accompanies the reader. The importance of serial correlation depends on the type of variable one wants to examine and on how the variable is used in the model. It cannot

be generalized to the application of NA methods as a whole. Incidentally, serial correlation also negates the assumptions of extreme value theory (EVT), with the effect of making the convergence much slower. Slow convergence actually suggests that NA methods should be used, making the reasoning circular and thus highlighting once again the complexity of the problem.

4.  Relatively large portions of the manuscript are dedicated to commenting specific works (sections 4.2, 5.3; 10 pages in total). Given that previous work, also by some of the authors (Serinaldi et al., 2020), and this work itself confirm that NA methods are formally correct, it is not fully clear how objections to specific works should affect NA methods in general.

Specific comments

5.  In section 3, the authors treat EVT as if it was the truth. Statistically it is, provided that the underlying assumptions are met. Among these the asymptotic assumption. In some relevant cases, convergence to the asymptote is (very) slow, such as the case of the powered exponential family of distributions. Notably, this is the case most relevant for precipitation, and precipitation is the main variable on which NA methods are confidently used (due to the relatively simpler relation with the underlying physics). In fact, in the case of precipitation tails from EVT are too heavy. This becomes clear when one tries to generate stochastic time series from a EVT distribution, and led to the development of a family of powered-exponential distributions for the generation of stochastic precipitation series (Papalexiou, 2022). These tails explain well the statistics of observed extremes, as shown by Marra et al. (2023) (more on this later). Overall, in that paper, we showed that GP tails from EVT and powered exponential tails from NA models can be indistinguishable, with the difference that the former are asymptotic distributions fitted to NA data. The message is once again that no model is perfect, and that different models may lead to similar answers, thus advancing our understanding of nature.

6.  Some concepts in Section 5.3.1 are misused. Montecarlo simulations (both in its standard term and in its Markov Chain variant) are numerical methods to compute integrals and expectation, sampling from a target distribution numerically and approximating the expectations via empirical average. However, the description provided by the authors is confused and, at least for what concern the different approaches to statistical inference (here the frequentist or classical paradigm and the

Bayesian one), wrong.

Montecarlo simulations in frequentist inference and Markov Chain Montecarlo (MCMC) in Bayesian inference target totally different objects. The authors correctly assess the role of Montecarlo simulations under a frequentist approach to statistics. Under this point of view there exists a true population's characteristic (or statistics, using the authors term) that is estimated (intrinsically with some uncertainty) from a finite sample. The variability of the estimator (and not of the parameter that the estimator is targeting) can be assessed in many ways, e.g. exploiting Montecarlo sampling to mimic the repeated sampling principle thus allowing to construct frequentist confidence sets. In Bayesian inference, instead, do not exist a 'true' parameter of the population as this is consider a random variable itself. Consistently with this, the posterior distribution of any unknown, which is often approximated via MCMC sampling is the target of inference. While posterior summaries like the posterior mean are common, they represent fundamentally different entities from frequentist estimators. In Bayesian inference, MCMC draws are used to construct credible sets, intrinsically different from the notion of frequentist confidence sets. The uncertainty that the posterior is describing is not the same uncertainty that the estimator variance in frequentist inference (obtained in any way, including Montecarlo sampling) is describing. Additionally, Bayesian model averaging is a well-known and successful concept that is not related to the summarization of the (MCMC approximated) posterior distribution of any kind.

Despite stemming from confusing arguments about basic concept of frequentist and Bayesian inference, the discussion starting from eq. (10) to the end of Section 5.3.1 is correct. However, it is a mere consequence of eq. (17) and deserves less space. Perhaps lines 453-485 can be removed and the subsequent text rearranged.

7. Section 5.3.2 is not clear. Specifically, I could not grasp whether the objection concerns (a) the average from the synthetic timeseries of the Montecarlo samples, or (b) the average in the MEV formulation. Are the authors claiming that the figure in Marani and Ignaccolo draws something different from what is claimed, or that the MEV framework is incorrect? The suggested changes in Fig. 7 indicate that we are in case (a). Should this be the case, the entire section 5.3 would be a direct comment to Marani and Ignaccolo (2015) that not necessarily pertain NA methods in general, but only the Montecarlo sampling in here. Should (b) be the case, it is not clear why section 5.3.1 is there and why all the distributions (not only MEV) change in figure 7. Even in this case, the comment would not pertain NA methods in general.

8. In section 6, the authors briefly comment on a paper of mine in which NA (Weibull) and asymptotic (GP) tails are compared for the case of precipitation. They quickly dismiss our study claiming that we used a low threshold "*out of its range of validity*". We reported results for threshold equal to the 95-th percentile for consistence with the Weibull model, but we clearly stated that "*Results derived from higher thresholds such as the 98-th percentile used by Serinaldi and Kilsby (2014) are qualitatively analogous but characterized by larger uncertainties*" (Marra et al., 2023). For reference, I report here the same as figure 3 in Marra et al. (2023) as it was obtained using a threshold equal to the 98-th percentile (Figure 1 below). As it can be seen, the instances in which GP provide too heavy or too light tails are even increased when using the 98-th percentile with respect to the 95-th percentile case (please refer to Marra et al., 2023). This is because in addition to theoretical convergence issues (what the authors focus on), there are important (practical) issues with stochastic (sampling) uncertainty.

9. The manuscript presents numerous self-citations.

10. Incidentally, as a user of NA methods, I never claimed they are 'superlative'. They are as good as other models are: they offer advantages in some situations and disadvantages in others.

[Figure]

*Figure 1: Same as figure 3a from Marra et al. (2023), but obtained using GP tails estimated with a threshold equal to the 98-th percentile*

---

## Author Comment (AC1)

**Non-asymptotic distributions of water extremes: Superlative or superfluous?**

**By F. Serinaldi, F. Lombardo, C.G. Kilsby**

**Submitted to *HESS**

**MS-NR: hess-2023-234**
* * *
**Reply Reply on RC1 (Anonymous Reviewer #1's report)**

(Note: In the text below, Referees' comments were copied verbatim in **black**.)

*Response* We thank the Reviewer for the interesting feedback. Please find below our response to raised concerns.

A classic choice in the statistical modelling of extremes is between (a) constructing a detailed model of an entire process, from which its extremal properties can be estimated, either analytically or more usually by numerical methods, or (b) direct modelling of the extremes themselves. If adequate reliable data are available and the investigator has sufficient time, then approach (a) allows information from other sources (such as physical models) to be included at the modelling stage and has the benefit that estimates of all quantities, including extremes, stem from a single overall model and therefore are consistent. However this approach is demanding of data and of time, and makes the implicit assumption that the details of the underlying process are relevant to the extremes. Approach (b) is less demanding of data and avoids detailed modelling by applying the classical theorems of extreme-value theory (EVT) to block maxima or threshold exceedances for the phenomenon of interest. Although originally developed for independent and identically distributed observations, these theorems have been shown to be robust to plausible types of dependence in the underlying data, and have been widely and generally successfully applied in environmental settings. They can be regarded as semiparametric models, in the sense that they do not depend heavily on the underlying process. A major concern is that they rely on limiting approximations (the GEV and GPD) that may fit data at observed levels satisfactorily but extrapolate poorly to unobserved levels. Such models provide a simple and direct empirical approach to modelling extremes of the underlying phenomenon but it may be a struggle to incorporate physical constraints or other background knowledge into them.

The paper under discussion can be viewed as a critique of a particular type (a) approach, namely metastatistical extreme-value (MEV) modelling, from the viewpoint of a classical type (b) approach, namely the fitting of GEV and GPD models to block maxima and threshold exceedances.

***Response*** We are grateful to the Reviewer for the thorough overview of statistical modelling of extremes. Anyway, we must clarify that in our work we do not compare asymptotic and non-asymptotic (NA) models of block maxima (BM) from the point of view of asymptotic models (the latter are indeed not even mentioned in the abstract). We contrast models of BM with the corresponding compound parent distributions. As clearly stated in the paper the key message is:

"*We discussed their redundancy and practical uselessness in real-world analysis. This apparently bold statement relies on very basic facts: (i) the distribution $F_Z$ of a process Z provides all information about any quantile or summary statistics (extreme or not); (ii) extreme value distributions $F_Y$ of BM corresponding to the parent process Z are just approximations of the tails of the distribution $F_Z$, and they have a role only if $F_Z$ is unknown; and (iii) NA distributions require the preliminary knowledge/estimation of $F_Z$; however, once $F_Z$ is known or fitted to data, NA distributions of BM are no longer needed, and their derivation is superfluous as $F_Z$ already provides all information. In this context, the use of asymptotic extreme value models is justified by the fact that they do not require the preliminary knowledge or estimate of $F_Z$ (under suitable conditions).*"

Asymptotic models are only mentioned in the paper to stress that their merit is to disconnect inference about the tails of the parent distribution from an accurate knowledge of the parent distribution itself (as mentioned by the Reviewer). Instead, NA models of BM require the preliminary inference of components (conditional marginals) that are sufficient to build the compound parent distribution. Therefore, NA models become useless as they provide just an approximation of the tails of a compound parent that is already completely defined. This is the message conveyed by Figs. 1 and 3 and Section 4.

Since we aim at conveying this message, we think that targeting "*much of the general text in the earlier sections for cuts*" is not a good idea, as these sections contain the epistemological justification of our criticism. This is also needed because we strongly believe that conceptual reasoning is crucial even in a modern era focused on massive data analysis and technicalities, because they must be supported by preliminary epistemological reasoning.

There are two main criticisms:

that papers proposing MEV have done so by application to and illustration on `real data', in which the true data-generating mechanism is unknown, which implies that it is impossible to compare the behaviour of different approaches under ideal conditions (when the target of inference is known);

that in any case the comparisons are incorrect, because of confusion over the target of inference (see Figure 7).  Here the point is more subtle, but it is summarised in equation (16) of the paper. The point here is that if one is estimating a quantile function $Q_\theta(p)=F^{-1}(p;\theta)$ that depends on an unknown parameter $\theta$ and one will estimate $\theta$ from a single sample

using an estimator $\hat\theta$, then the estimator of $Q_\theta(p)$ is $Q_{\hat\theta}(p)$, whose properties should be assessed over repeated sampling using independent replicates $Q_{\hat\theta_1}(p), \ldots, Q_{\hat\theta_S}(p)$ based on $S$ samples leading to estimates $\hat\theta_1,\ldots, \hat\theta_S$.    The average of these estimates would be $S^{-1} \sum_{s=1}^S Q_{\hat\theta_s}(p)$, i.e., the right-hand side of (16), rather than $Q_{\bar \hat\theta}(p)$ (the left-hand side of (16)), where $\bar\hat\theta$ is the average of the parameter estimates for the $S$ samples.  The paper under discussion illustrates the difference via the left- and right-hand panels of Figure 7.

*Response* Thanks again for this interesting comment. However, as mentioned above, our main concern comes from introducing NA models (not only MEV, but also our own models in Eq. 4) neglecting the epistemological justification of EVT (asymptotic) models.

In real-life problems, we mainly need the probability of values taken by processes such as discharge, precipitation intensity/depth, water level, etc. We do not need or look for the probability BM or POT of such processes. We focus on BM or POT for convenience, as these sub-processes can conveniently be described by a couple of distributions that do not require the knowledge of the parent process (under suitable conditions).

Since NA models of BM do not exhibit this disconnection, they automatically lose the only practical advantage of using "sub-optimal" BM and POT processes.

All the technical inconsistencies discussed throughout the paper are just a consequence of missing these epistemological concepts, reducing the development of new models to a mechanistic exercise focusing on "how" to do that, but missing the fundamental preliminary step, i.e. "why" we develop new models! This is the meaning of (the title of) Section 3: "Modeling extreme values: asking 'why' before looking for 'how'".

The literature on NA models reveals that these concepts are widely overlooked and, perhaps, need to be recalled as we did in the manuscript.

(Though the discussion at lines 532-535 leaves it unclear how the `median GEV/Gumbel' curves are computed — the median for each $p$, giving a result that would not corresponding to any single quantile function, or what?  And if the median, why not the mean?).

*Response*

Median GEV/Gumbel probability and quantile functions are obtained by Eq. 14 and 15.

We state "*Therefore, to be consistent with EVT, the ensemble of GEV and Gumbel distributions should be summarized using a transformation, such as the median, that retains the expected GEV/Gumbel shape.*" In this respect, we devote the preliminary sub-section 5.3.1 to explain why the mean provides biased summary of MC ensembles, whereas rank-based summary statistics, such as the median, are insensitive to non-linear transformations linking parameters, $F_Z$, and $Z$, and are therefore more appropriate to preserve the shape of the distribution. To further explore such concepts, a graphical representation is shown in figure below (with self-explanatory labels); it

refs to GEV distribution with shape parameter equal to 0.1, sample size equal to 50, and 3000 Monte Carlo replications to assess sampling uncertainty.

[Figure]

Moreover, we introduced Section 5.3.1 by explicitly stating that "*We anticipate that the foregoing discrepancies depend on the misuse of methods used to summarize multi-model ensembles. Thus, before describing Monte Carlo experiments and their outcome, we need to recall some theoretical concepts that are required to correctly interpret numerical results.*"

Therefore, the text already cares about the reader's understanding of the discussion. As mentioned above, specific sentences in the text are consistent with the premises reported in the introducing sections.

Both of these criticisms seem to me to be correct, and they should in my view embarrass the reviewers of the original MEV papers and the journals that published them.

*Response* We believe that the objective statement by an independent Reviewer about the correctness of our criticisms is fitting to the task for the Authors of this paper, who have honed their research over the years to the very aims of mapping out and understanding theoretical consistency in analysis of geophysical data, without getting much consensus for that. Therefore, the Reviewer has our genuine thanks for providing us with such an important comment in acknowledgement of our work. Other than that, we agree with the Reviewer that "new" methods could sometimes be uncritically accepted in scientific journals and then routinely applied by the scientific community without double-checking their theoretical basis, taking for granted that they are conceptually/formally correct just because they are published once somewhere.

I found the paper to be quite poorly written, to the point of unclarity in numerous places, including lines 405 ('the spreader …'?), 429 (' …, or better, …'?) or 524 (what is a predictive quantile function of a predictive quantile function?), and with many minor errors. Examples of the latter are that (i) the Beirlant et al book cited at line 18 was published in 2004, not 2006, and (ii)

stating on line 476 that the distribution of an order statistic is beta is incorrect — the beta distribution represents variables on a finite interval, and clearly this does not apply to order statistics from, say, a Gaussian sample (did the authors mean that the distribution of an order statistic can be represented _using_ that of a beta random variable?), (iii) equation (17), the left-hand side of which is a function of $z$, while the right-hand side is a number (as the expectation of $Z$ is a constant), and (iv) at line 461, where results from a simulation study are `eventually used to build confidence intervals' — but in a simulation study the truth is known, so confidence intervals are not needed — as a confidence interval is based on a single sample, we have to guess that the authors mean that their $S$ return level estimates will be used to compute quantiles of a distribution.  The paper is full of inaccuracies of this sort, so the reader is continually wondering `is that correct?' and concluding `not quite'; this does not give confidence in the main results.   It is the role of the authors to produce a well-crafted article, not that of a reviewer, so I will not give more examples (it would take many pages to list them all), but generally I found the writing to be unclear, long-winded, and in need of a careful review by a native English-speaker (see, e.g., line 514).  Reducing the paper radically by revising and trimming the text throughout would improve it. I would also target much of the general text in the earlier sections for cuts, since it is mostly not germane to the criticism of the MEV work.  A 15-page paper in the current format would make the main points more clearly and should be more readable.

*Response* We will double-check language and presentation. Concerning the specific points:

- *"predictive quantile function"*: the full sentence states that *"the predictive quantile functions (over S samples) of the predictive quantile functions (over $n_Y$ samples) associated to MEV structure"*.

  Predictive quantile functions are just ensemble averages: Figure 7 just shows the ensemble averages over $S$ samples of the ensemble averages over the $n_Y$ quantile functions contributing to the MEV quantile function. In other words, there are two levels of compounding (let us say hierarchy): the first one is related to the derivation of MEV (which is itself a predictive distribution integrating the inter-block variability of parameters), while the second one is related to the derivation of the predictive version of MEV, integrating (averaging) over the $S$ MEV functions.

  Thus, the curves in Figure 7 are ensembles of ensembles, where the average is taken firstly over $n_Y$ parent quantile functions and therefore over $S$ MEV quantile functions. This is formalized in L. 525, where we state *"In more detail*, $z_{p,\bar{Q}} \cong E_S[E_{\Omega_{\theta_S}}[F_{WEI}^{l}{}^{-1}(p|\theta_S)]]...$"

(i)   We will fix the typo in the BibTex entry of Beirlant et al., thanks.

(ii)  We understand Reviewer's concern and we will clarify this point in the revised text as follows.  Under i.i.d. assumption, order statistics have a binomial distribution (which is equivalent to a beta) in the sense described by Equation 1 (see also David and Nagarajah 2003, pp.9-10). In other words, the distribution of the order statistics is a beta distribution of the variable $F_z(Z)$, or equivalently a so-called beta-extended distribution of $Z$ (Eugene et al. 2002), which is also known as generalized beta-G

distribution, where "G" denotes generalized classes, such as exponentiated-G or Kumaraswamy-G (e.g., Tahir and Cordeiro, 2016).

Therefore, from Eq. 1 and the expression reported in L.476, it should be clear that the range of the distribution is not bounded.

(iii) Thanks, we will double-check notation and fix these typos.

(iv) We believe a sentence cannot be extrapolated from the context. We state: "*Monte Carlo simulations are usually used to study the uncertainty affecting estimates based on finite-size samples (that provide incomplete information about the underlying process) or to approximate population distributions (or statistics) when mathematical closed-form expressions are not available. Examples of these applications are the experiments reported in Sections 5.1 and 5.2… In all cases, the primary output of Monte Carlo simulations is a set of parameters identifying a set of models (multi-model ensemble) that is then used to estimate the target statistics of interest. For example, simulations of S finite-size samples in Sections 5.1 and 5.2 are used to fit a set of S GEV distributions. These are then used to calculate a set of S 100-year return levels, which are eventually used to build confidence intervals*"

The sentence refers to the use of MC to build confidence intervals describing the uncertainty of estimates from finite-size samples.

To summarize, we will fix the above issues by double-checking notation and typos and adding some minor details. Anyway, we also think that these issues are far from being sufficient to call into question the overall content of the paper.

Concerning the length of the paper, we believe the message of the first part of the paper is very important despite its simplicity and iteration throughout the text. We do believe that in a shorter version such message could be easily neglected or misinterpreted.

We also highlight the content from Section 4.1.1 to 5.2 (i.e., L. 178-437), where we discuss key points such as the interpretation of the relationship between parent models and models of BM, the effect of serial correlation, and provide examples by simulations and data re-analysis. The content of the paper goes far beyond Sections 3 and 5.3, we will stress this in the revised manuscript.

**References**

Eugene N., Lee C., Famoye F. (2002) Beta-Normal distribution and its applications, Communications in Statistics - Theory and Methods, 31:4, 497-512

Tahir, M.H., Cordeiro, G.M. Compounding of distributions: a survey and new generalized classes. J Stat Distrib App 3, 13 (2016)

---

## Author Comment (AC2)

**Non-asymptotic distributions of water extremes: Superlative or superfluous?**

**By F. Serinaldi, F. Lombardo, C.G. Kilsby**

**Submitted to *HESS***

*MS-NR: hess-2023-234*
* * *
**Reply on RC2 (Dr. F. Marra's report)**

(Note: In the text below, Referees' comments were copied verbatim in **black**.)

We thank the Reviewer for the constructive feedback. In the following, we provide point-by-point responses in **blue**.

This manuscript critics the use of non-asymptotic (NA) distributions of block maxima. The authors bring two main arguments to support their critic. The first is that NA require knowledge of the parent distribution and that, when this is known, there would be no need for deriving distributions of block maxima. The second is that the presence of serial correlation in the observations would decrease the potential advantage of NA. The manuscript then follows with some targeted comments to specific studies. The paper addresses a relevant topic, but the manuscript falls short at supporting its main conclusions. This is not due to technical errors, rather to a narrow (mono-disciplinary) vision of the problems at hand (see main comments) and to an erroneous generalization of case-specific objections. Since all statistical models present advantages and disadvantages, which depend on how much the underlying assumptions are met/not met (or, to quote the statistician George Box: "all models are wrong, but some are useful"), I believe that the presented critics cannot be generalized to NA methods as a whole. Rather, they should be targeted to highlight specific aspects of NA methods that need attention and/or the specific NA approaches that need attention.

Considering the main comments below, I believe the manuscript should be deeply revised before reconsideration. I am not sure this can be handled within a major revision, because the take-home messages would need important adjustments. The title should be revised to be more pertinent with the actual outcomes. This also pertains the title of some sections/subsections which border disrespectfulness (e.g., section 3, 4, 5). The manuscript is long and contains numerous repetitions. It includes unclear and/or incorrect reasoning in some sections, which prevent from fully

understanding some parts (see comments 6 and 7 below). It could be halved in length without changing the message.

All the references in this document can be found in the preprint of the manuscript.

Given my lack of specific expertise, comment 6 was written with the help of a colleague expert in Bayesian statistics.

I hope my comments are helpful.

Kind regards,

Francesco Marra

**Response** We thank the Reviewer for the time devoted to our paper. We are also grateful to the Reviewer who acknowledges that the paper is technically correct, thus he necessarily agrees that our conclusions are also technically correct. As discussed below, we believe that such conclusions are also fully general, as they apply to any NA model, including the ones introduced by the Authors of this paper (see e.g., Serinaldi et al., 2020b and Lombardo et al., 2019).

We agree that some concepts are sometimes repeated in the paper text, because we understand that they have been usually neglected in a huge amount of the literature dealing with NA models of block maxima (BM). Thus, we decided to follow the statement by the renowned physicist Arthur Leonard Schawlow: "Anything worth doing is worth doing twice", or even repeated more times when it comes to statistical analyses.

Main comments

1. My main comment turns out to be a citation from the manuscript itself (line 694): "models and methods should be thought and used in the right context and for suitable purposes". This sentence in the conclusions contradicts several of the arguments presented in the manuscript. The authors proceed for 30 pages (pages 1-30) repeatedly claiming that NA methods are 'superfluous'. Only in lines 685-694 they contradict this argument stating that what they report in the manuscript "does not mean that they are not useful at all". They then proceed listing what, in their view, are potentially useful applications of NA approaches and finish off with the citation I started with. I may add that the reported list of 'useful' applications is limited by the imagination of the authors (as shown by the sole presence of self-references in here) and, mostly, by the perspective they adopt. In this, it seems they forget the gap that there exists between theory and practice, between advancements in theory and practical use of extreme value distributions of any kind by hydrologists, risk modelers and end users in general.

**Response** We kindly invite the Reviewer to read our paper more carefully. Indeed, it seems here that the Reviewer has extrapolated the meaning of some incomplete sentences out of their context giving rise to misunderstandings and wrong interpretations of the Authors' statements.

Actually, highlighting the difference between theory and practice is the very aim of our paper. Indeed, throughout the text, we always and purposely use periphrases like "*little usefulness for*

*practical applications*", "*usefulness of NA models in practical applications*", "*the problems concerning the use of NA models of BM for practical applications*", "*call into question the practical use and usefulness of NA*", etc., etc., whereas L685-694 refer to usefulness of NA models in *theoretical* context, as should be obvious to everyone.

The Reviewer is pointed to the distinction between practical and theoretical usage of NA models of BM, which is anticipated in L. 174-176, where we clearly state "*This explains why NA have not received much attention and why the recently proposed compound NA models are of little **practical** usefulness, if any. Their usefulness is mainly **theoretical**, as they help explain the inherent differences between parent processes Z and BM processes Y, thus avoiding misconceptions and misinterpretation of different model outputs (see Serinaldi et al., 2020b).*"

Thus, it is quite evident that there is no contradiction at all.

Furthermore, Reviewer's remark does not consider the key point, that is, the argument supporting our conclusion about the practical uselessness of NA models of BM: "*NA models of BM imply the preliminary definition of their conditional parent distributions, which explicitly appears in their expression. However, when such conditional parent distributions are known or estimated also the unconditional parent distribution is readily available, and the corresponding NA distribution of BM is no longer needed, as it is just an approximation of the upper tail of the parent*".

In fact, several applications of NA methods follow the directions accepted as 'useful' by the authors, and other applications of NA methods follow directions that are useful, although not within the directions imagined by the authors.

*Response* We disagree here with the Reviewer, as he seems to confuse practice with theory, namely theoretical and applied statistics. A clear understanding of such difference can be derived by reading (and comparing) for example Shao (2003; Mathematical Statistics) and Kottegoda and Rosso (2008; Applied Statistics for Civil and Environmental Engineers), among others.

One notable example is the connection between physical processes and statistics, which can only exist in a NA model, given that real world physics is not asymptotic. It is my believe that physical processes should direct the statistics we use. The physics of the processes we are dealing with is not asymptotic.

At this concern I must cite again the authors (lines 623- 624): "Historically, the main scientific progresses occurred when some one called into question widely accepted mainstream theories using arguments more solid than those of the superseded theories". It almost seems we think alike on this point, although with different concepts for 'mainstream'.

*Response* We are sure we think alike with the Reviewer on several points to such an extent that he previously stated that our paper is technically correct. In other words, he says our work is correct according to a strict interpretation of the rules. That means that the Reviewer agrees on all the scientific rules we have extensively pointed out in the paper, where we also showed that such rules are unclear, misinterpreted or even neglected in a huge part of the literature. Then, we are

more confident in our vision that we can claim we are providing a valuable contribution to the scientific community. Other than this, we are afraid we did not understand the first sentence of Reviewer's remark, because:

- "Physical" (natural, or manmade) processes are what we observe around us.
- Physics (intended as a body of theories) and statistics are just modeling frameworks.
- Concepts like "asymptotic" or "non-asymptotic", "deterministic" or "non-deterministic", can only refer to models not to "*real world physics*", whatever it means.
- Observation records are yet another thing, and they have finite size.

If we say that "*real world physics is not asymptotic*", we can also say, for instance, that "real world physics is not *deterministic*": should we discard Newton's classic mechanics because "*real world physics*" is not uncertainty-free?

As stated by Morrison (2008) "*The next hurdle* [to get over in undergraduate mathematics] *is the differences among observed reality, mathematical models, and computational realizations of mathematical models. Even a lot of accomplished scientists are not clear on these points... learning to cope with three things makes up the basics of a liberal scientific education: facts, abstractions, and the comparison of facts with abstractions... Understanding and ultimately research occurs only when facts are reduced to abstraction, the abstractions manipulated to make predictions, and the prediction compared with new facts*".

Nowadays, it seems that there is a big confusion about basic epistemological and semiotic concepts, which are fundamental to make meaningful statements.

2. One argument is that NA methods require the knowledge of the parent distribution and that when this is known, there would be no need for deriving a block maxima (BM) distribution. This is technically true but seems to neglect situations in which a BM distribution is helpful (even though, I agree, not technically essential). Some examples: empirical comparison with observations of BM only; fair comparison with estimates from EVT distributions; providing information that practitioners can use without changing habits. Doing these directly from the compound parent, although possible, would be troublesome and possibly confusing for non-experts.

*Response* Again thanks to the Reviewer who thinks that our arguments are technically correct. Reviewer's fairness is indeed greatly appreciated. Concerning the situations where models of BM (asymptotic or not) would be helpful, we refer to our paper's Section 4, which already addresses Reviewer's remarks:

1) Rescaling compound parent distributions has a degree of complexity that is always less or equal to deriving the corresponding NA models for BM (see Section 4.1.1).
2) Why should one compare (superpose) the distribution of a given process (e.g. streamflow) with the empirical (or a theoretical) distribution of annual maxima in a practical situation? Once we know the probability of (non)exceedance of a given streamflow value, this is all we need, indeed:
    a. If BM (annual maxima) are the only available data, NA models cannot be built.

b. If we have enough data to build NA compound parent models, BM models are irrelevant.

BM datasets are not of interest *per se* in any real-world application. They are only functional to rebuild the upper tail of the distribution of the parent process via distributions that hopefully do not require the (detailed) knowledge of the parent distribution (under suitable conditions). Other than that, BMs have no special purpose in practical applications, engineering design, management, etc.

3) "*fair comparison with estimates from EVT distributions*": Figures in the paper show fair comparisons among NA models of BM, NA parent models, and EVT models. More importantly, from a practical standpoint, once the parent model is available (and assumed to be reliable), any other model of BM (asymptotic or not) is just an approximation of its upper tail: as an approximation, it is always less accurate/correct than the parent models. Why should we build and compare two models of BM, when we already have a distribution that is superior by construction? Models of BM have no longer place once we decide to build/recover (compound) $F_Z$.

4) "*providing information that practitioners can use without changing habits*": using parent distribution has at most the same degree of difficulty as using a POT distribution (such as the classic GP) in terms of derivation of return period or other summary statistics. Moreover, how can a practitioner be more comfortable with models that are more convoluted than their parent models?

5) In our experience, "non-expert" and "practitioners" should be trained to properly use models and methods rather than providing them with more and more convoluted models that they do not know/understand and likely misuse due to apparent user-friendliness. On the other hand, if practitioners are well trained and can understand the nature and structure of compound NA models of BM, they will recognize that the parent models are the most straightforward option.

3. The issue with serial correlation is important and could affect some applications of NA methods. I believe future NA applications (either for block maxima or directly from the parent distribution) should keep this in mind. In this, the paper is a relevant addition to the literature. Still, it falls short at supporting the adjective 'superfluous' that accompanies the reader. The importance of serial correlation depends on the type of variable one wants to examine and on how the variable is used in the model. It cannot be generalized to the application of NA methods as a whole. Incidentally, serial correlation also negates the assumptions of extreme value theory (EVT), with the effect of making the convergence much slower. Slow convergence actually suggests that NA methods should be used, making the reasoning circular and thus highlighting once again the complexity of the problem.

**Response** We kindly reply to the Reviewer with the following points that are already included in our manuscript:

1) We do not vaguely talk about serial correlation and dependence: we specifically show that "*when declustering procedures are used to remove autocorrelation characterizing hydro-climatic records, NA distributions of BM devised for independent data are strongly biased even if the original process exhibits low/moderate autocorrelation. On the other hand, NA distributions of BM accounting for autocorrelation are less biased but still of little practical usefulness*" because they are yet approximations of the already available compound parent distributions. The NA models of BM are redundant in any condition (dependence, independence, stationarity, non-stationarity, or anything else).

2) EVT can take some kind of dependence into account by e.g. extremal index. In other cases, such as the presence of strong dependence, EVT just says that the asymptotes can be different from GEV/GP.

3) We do not say anywhere that NA methods should not be used: we say that NA models of BM are redundant in practical applications as they just approximate the upper tail of the already available NA parent distributions (Figs. 1 and 3), and both are biased under dependence and independence (Figs. 4 and 5).

   We do not even contrast asymptotic and NA distributions of BM: we show that the NA models of BM miss the main point that justifies the use of BM models, that is, no need of information about the precise nature of the parent distributions.

4) There is no circular reasoning affecting asymptotic and NA models: if we have only BM, NA models cannot be defined. If we have complete data sets, we can build (compound) parent distributions and we do not need any NA model of BM. Alternatively, we can use EVT (asymptotic) models if we do not want (or cannot) fit a suitable parent model for some reason.
   Circular reasoning only affects NA models of BM ("children") when compared to their compound parent ("parent"). Indeed, we generate the "parents" to give birth to "children", then we use the "children" to recover part of the "DNA" of the parents, which is already known! This is circular reasoning.
   Conversely, "dependence" (and slow convergence) is just a technicality that can be addressed asymptotically or not. Moreover, EVT goes far beyond GEV and GP distributions. Circular reasoning concerns logical arguments rather than technicalities.

We believe the relevance and importance of such concepts described above justifies their various repetitions throughout the text of our manuscript.

4. Relatively large portions of the manuscript are dedicated to commenting specific works (sections 4.2, 5.3; 10 pages in total). Given that previous work, also by some of the authors (Serinaldi et al., 2020), and this work itself confirm that NA methods are formally correct, it is not fully clear how objections to specific works should affect NA methods in general.

*Response* The Reviewer is kindly pointed to Section 4.2 of our manuscript, which clearly shows the redundancy and practical uselessness of NA models of BM: "*being formally correct/incorrect*", "*being biased/unbiased*", and "*being useful/useless*" are different things. For example:

- Under serial dependence, the classical estimator of the correlation coefficient is (i) formally correct, (ii) biased, and (iii) only partially useful (as there are better estimators of linear dependence).
- NA models of BM are (i) formally correct, (ii) biased, and (iii) practically useless in real world applications (data analysis).

On the other hand, our Section 5.3 shows that the simulations used in the literature to support MEV models are formally incorrect as they confuse expected quantile functions and expected probability functions.

While we refer to specific papers, our remarks are fully general: all NA models are redundant when compared with the corresponding (and known) NA parent models (Section 4.2), and all NA models (parent and BM) are biased because of their compound nature (Sections 5.1, 5.2 and 5.3).

We are sure that the Reviewer is fully aware of the difference between general conclusions and case-specific conclusions. If he thinks that our statements are not general and/or not supported by our analyses, we will be glad to discuss possible counter examples that he can provide if he wants.

Specific comments

5. In section 3, the authors treat EVT as if it was the truth. Statistically it is, provided that the underlying assumptions are met. Among these the asymptotic assumption. In some relevant cases, convergence to the asymptote is (very) slow, such as the case of the powered exponential family of distributions. Notably, this is the case most relevant for precipitation, and precipitation is the main variable on which NA methods are confidently used (due to the relatively simpler relation with the underlying physics). In fact, in the case of precipitation tails from EVT are too heavy. This becomes clear when one tries to generate stochastic time series from a EVT distribution, and led to the development of a family of powered-exponential distributions for the generation of stochastic precipitation series (Papalexiou, 2022). These tails explain well the statistics of observed extremes, as shown by Marra et al. (2023) (more on this later). Overall, in that paper, we showed that GP tails from EVT and powered exponential tails from NA models can be indistinguishable, with the difference that the former are asymptotic distributions fitted to NA data. The message is once again that no model is perfect, and that different models may lead to similar answers, thus advancing our understanding of nature.

***Response*** We regret to say we disagree with the Reviewer here for the following two reason:

- Section 3 does not treat EVT as the truth, whatever "truth" means.
- "*Statistically it is*"? What does it mean that EVT is a statistical truth?

We do not think that such comment is related to the content of Section 3 of our paper. Let us summarize Section 3. Its title is "*Modeling extreme values: asking 'why' before looking for 'how'*" because we noted that the literature on NA models of BM is so focused on convoluted transformations of the parent distributions $F_Z$ that it seems to miss the key point: for a process $Z$,

$F_Z$ already provides all information about the probability of every quantile (extreme or not). We do not need any other distribution, which is necessarily less informative of $F_Z$.

Even the Reviewer's remark seems to talk about "how" (convergence, Weibull distributions, simulations, etc.), and it does not focus on "why" we develop such models, which is instead the topic of Section 3.

Distributions of BM have been studied as one hopes to get insights into the tails of $F_Z$ **when $F_Z$ is not available** for some reason (e.g., lack of data, or difficulty to reliably identify $F_Z$).

**When $F_Z$ is available**, we do not need any distributions of BM (either asymptotic or non-asymptotic) to define the probability of any quantile. Indeed, in real world applications, we need the probability of $z$ (discharge, rainfall, etc.), not the probability of BMs (streamflow AM, or rainfall AM).

*"One might wonder why we should be interested in an asymptotic distribution of Y when the exact distribution, which is given by $F_Y(z) = F^m_Z (z)$, where $F_Z$ is the c.d.f. [cumulative distribution function] sampled from, is known. **The hope is that we will find an asymptotic distribution which does not depend on the sampled c.d.f. $F_Z$.**"* (Mood et al., 1974, p. 258).

To summarize:

1) The process of interest (rainfall, streamflow, etc.) is $Z$.
2) The probability of any quantile is described by $F_Z$.
3) When $F_Z$ is known, we do not need anything else to calculate the probability of any $z$.
4) When $F_Z$ is unknown:
    a. Asymptotic models of BM provide an approximation of the tails of $F_Z$ that **does not require** the (precise) knowledge of $F_Z$.
    b. NA models of BM cannot be derived as they **require** the preliminary knowledge of $F_Z$, which explicitly enters in their expression.

Then, we highlight once again that searching for an approximation of the upper tail of $F_Z$ makes no sense if the whole $F_Z$ is already known: why should one build an inferior approximate model of a sub-process, when one already has a superior model describing the whole process?

Supporters of NA models of BM always contrast these models with asymptotic models, without recognizing that the true competitors of NA models of BM are the parent distributions that need to be preliminarily identified for the derivation of NA models of BM. And parent models are always superior to any model of BM as they describe the whole state space of process $Z$, whereas models of BM (asymptotic or not) describe just a subset.

For clarity, we further summarize the key points graphically in the figure below.

[Figure]

(Compound) parent distributions of Z (orange line) are more informative than any and every asymptotic or non-asymptotic model of BM (grey line) as they describe the whole state space.

Asymptotic or non-asymptotic models of BM (grey line) are just approximations of the upper tails of parent models (orange line).

NA models of BM are redundant, as they require the preliminary identification of the parent distributions, which are more informative than their surrogate NA models of BM.

The "querelle" between supporters of asymptotic models and NA models of BM is ill-posed: the competitors of NA models of BM are not the EVT models, but the more informative parent distributions required for their derivation (and appearing in their expression).

6. Some concepts in Section 5.3.1 are misused. Montecarlo simulations (both in its standard term and in its Markov Chain variant) are numerical methods to compute integrals and expectation, sampling from a target distribution numerically and approximating the expectations via empirical average. However, the description provided by the authors is confused and, at least for what concern the different approaches to statistical inference (here the frequentist or classical paradigm and the Bayesian one), wrong. Montecarlo simulations in frequentist inference and Markov Chain Montecarlo (MCMC) in Bayesian inference target totally different objects. The authors correctly assess the role of Montecarlo simulations under a frequentist approach to statistics. Under this point of view there exists a true population's characteristic (or statistics, using the authors term) that is estimated (intrinsically with some uncertainty) from a finite sample. The variability of the estimator (and not of the parameter that the estimator is targeting) can be assessed in many ways, e.g. exploiting Montecarlo sampling to mimic the repeated sampling principle thus allowing to construct frequentist confidence sets. In Bayesian inference, instead, do not exist a 'true' parameter of the population as this is consider a random variable itself. Consistently with this, the posterior distribution of any unknown, which is often approximated via MCMC sampling is the target of inference. While posterior summaries like the posterior mean are common, they represent fundamentally different entities from frequentist estimators. In Bayesian inference, MCMC draws are used to construct credible sets, intrinsically different from the notion of frequentist confidence sets. The uncertainty that the posterior is describing is not the same uncertainty that the estimator variance in frequentist inference (obtained in any way, including Montecarlo sampling) is describing.

*Response* We thank the Bayesian statistician involved by the Reviewer for his/her overview about the scope of MC simulations, and the difference between frequentist confidence intervals and Bayesian credible intervals. We are fully aware of such concepts, being already familiar with the explanations provided by e.g. Nicholas Metropolis, Arianna and Marshall Rosenbluth, Stanisław Ulam, and Edward Teller in their original papers on MC, or Bernardo and Smith (2000), Gelman et al. (2003), or Robert (2007) in their Bayesian books, etc., being left alone the original De Finetti's works on subjective probability.

However, we must stress here that section 5.3.1 and the whole paper have nothing to do with Bayesianism or the "*46656 varieties of Bayesians*" (Good, 1971, Am Stat 25:62–63).

Our paper does not report any credible interval or Bayesian analysis. In the first paragraph of Section 5.3.1, we just state that simulation methods have several applications, and one of them is "*to obtain posterior distributions of model parameters with unknown mathematical form*", that is, when closed form of posterior distribution is not available, which is the most common case in data analysis.

Summarizing a multi-model ensemble is a general problem that is fully independent of the inferential strategy, as multiple models can result from sampling uncertainty analysis in frequentist fashion, from posterior distributions in Bayesian inference, or just from multiple physical models (with different model structure) without involving any statistical inference. The third paragraph in Section 5.3.1 states that the problem of summarizing multiple models is well known for instance in Bayesian literature just because the typical output is a set of models corresponding to parameter sets usually sampled via MCMC. And this has nothing to do with the difference between credible intervals and confidence intervals, which should be indeed well known to anyone who uses applied statistics to "play" with data.

Finally, the third and last entry of the term "Bayesian" is in L. 646, where we discuss "predictive distribution". However, also in this case, Bayesian inference/theorem does not apply whatsoever, as predictive distributions are just an application of the total probability theorem and marginalization.

In this respect, we endorse the following statement by Christakos (2010): "*when an investigator was asked if he is a "Bayesian" or a "non-Bayesian," he responded that he is an "opportunist," meaning that he would use whatever approach works best for the given in situ conditions*".

Additionally, Bayesian model averaging is a well-known and successful concept that is not related to the summarization of the (MCMC approximated) posterior distribution of any kind.

**Response** Indeed, we do not apply and do not even mention "Bayesian model averaging" anywhere in the paper.

Despite stemming from confusing arguments about basic concept of frequentist and Bayesian inference, the discussion starting from eq. (10) to the end of Section 5.3.1 is correct. However, it is a mere consequence of eq. (17) and deserves less space. Perhaps lines 453-485 can be removed and the subsequent text rearranged.

**Response** Thanks indeed to the Reviewer and his Bayesian colleague for stressing once again the correctness of our paper. We believe indeed that the discussion after Equation 10 is correct because its premises are correct, as they are fully general, and do not refer and are not limited to frequentist or Bayesian inference: they refer to how one can summarize multiple models, and this has nothing to do with the specific inferential procedure (often, involved models are not even statistical).

Lines 453-485 are necessary to introduce predictive and median quantile/probability functions that play a key role in the interpretation of results in Section 5.3.2 (as explicitly stated in L. 450-451).

Clearly, Section 5.3.1 must be read in the context as clearly recommended in L. 449-451.

7. Section 5.3.2 is not clear. Specifically, I could not grasp whether the objection concerns (a) the average from the synthetic timeseries of the Montecarlo samples, or (b) the average in the MEV formulation. Are the authors claiming that the figure in Marani and Ignaccolo draws something different from what is claimed, or that the MEV framework is incorrect? The suggested changes in Fig. 7 indicate that we are in case (a). Should this be the case, the entire section 5.3 would be a direct comment to Marani and Ignaccolo (2015) that not necessarily pertain NA methods in general, but only the Montecarlo sampling in here. Should (b) be the case, it is not clear why section 5.3.1 is there and why all the distributions (not only MEV) change in figure 7. Even in this case, the comment would not pertain NA methods in general.

*Response* The justification of the analysis in Section 5.3.2 is explicitly and clearly stated at the beginning of Section 5.3: "*Therefore, we re-run Monte Carlo simulations described by Marani and Ignaccolo (2015) to understand the reason of such a disagreement* [with simulations in Section 5.1 (reproducing those of Marra et al. (2018))]. *We anticipate that the foregoing discrepancies depend on the misuse of methods used to summarize multi-model ensembles. Thus, before describing Monte Carlo experiments and their outcome, we need to recall some theoretical concepts that are required to correctly interpret numerical results.*"

The message of results in Section 5.3.2 is indeed very simple:

- NA models (BM and parent) are biased thus confirming results in sections 5.1 and 5.2.

- These results contrast with those of Marani and Ignaccolo (2015), which are affected by incorrect use of multi-model averaging over $S$ and $\Omega_{\theta_S}$.

- Since results reported by Marani and Ignaccolo (2015), including apparent lack of bias, are routinely used to justify the goodness of NA models of BM (due to supposed better performance with respect to EVT models), Section 5.3.2 shows that such arguments are not valid.

To conclude, our concerns refer to both options (a) and (b), which are not mutually exclusive, even though we must rephrase them for the sake of correctness. We state that the figures in Marani and Ignaccolo show something different from what is claimed, and that the MEV framework and any NA model (BM or parent) is biased (… not "incorrect"). Indeed, in statistical modelling there is no "free lunch": what we gain in reduced variance, we lose in increased bias, and vice versa. However, NA models are routinely described in the literature missing their bias, which disappeared in the incorrect diagrams reported in a paper that is usually cited as a starting point for these NA models of BM (neglecting numerical errors, lack of correspondence between figures and text, etc.).

8. In section 6, the authors briefly comment on a paper of mine in which NA (Weibull) and asymptotic (GP) tails are compared for the case of precipitation. They quickly dismiss our study claiming that we used a low threshold "out of its range of validity". We reported results for threshold equal to the 95-th percentile for consistence with the Weibull model, but we clearly stated that "Results derived from higher thresholds such as the 98-th percentile used by Serinaldi and Kilsby (2014) are qualitatively analogous but characterized by larger uncertainties" (Marra et al., 2023). For reference, I report here the same as figure 3 in Marra et al. (2023) as it was obtained using a threshold equal to the 98-th percentile (Figure 1 below). As it can be seen, the instances in which GP provide too heavy or too light tails are even increased when using the 98-th percentile with respect to the 95-th percentile case (please refer to Marra et al., 2023). This is because in addition to theoretical convergence issues (what the authors focus on), there are important (practical) issues with stochastic (sampling) uncertainty.

***Response*** We disagree with the Reviewer here, because the foregoing remark starts from premises or statements that are attributed to our paper even though they do not appear anywhere.

Furthermore, we do not dismiss any paper: we call into question the interpretation of Fig. 5 (not Fig. 3!) in Marra et al. (2023), which has practical consequences as clearly stated in L. 571-592. The Reviewer interpreted that figure (reported below for convenience) as follows:

"*the errors for maxima sampled from GP tails strongly depend on the left-censoring threshold and tend to be too heavy-tailed for $\vartheta \leq 0.90$. The accuracy of GP tails in reproducing the statistics of observed maxima is comparable to the one of Weibull tails only for thresholds $\vartheta > 0.9$. Second, GP\* tails estimated from synthetic Weibull-distributed data, are virtually indistinguishable from the GP tails estimated from real observations (dashed blue). As predicted by EVT, GP tails tend to provide similar estimates upon asymptotic conditions (here represented by $\vartheta_{GP} = 0.95$; see also Serinaldi and Kilsby, 2014) and the difference in L-moment ratios between non-asymptotic Weibull tails and GP tails decreases with increasing threshold (Fig. 5c). Crucially, the difference between L-moment ratios of annual maxima emerging from GP and GP\* tails (dashed) are virtually indistinguishable also for high thresholds such as $\vartheta = 0.95$, and smaller than the differences between L-moment ratios of annual maxima emerging from GP and Weibull tails (Fig. 5c). Estimating GP tails from observations is equivalent to estimating GP tails from Weibull data.*"

[Figure]

[Figure]

[Figure]

**Fig. 5.** Error in L-skewness (a) and L-kurtosis (b) of annual maxima estimated from MC samples of $10^3$ years of non-asymptotic Weibull tails (WEI, red) and GP tails (blue) with respect to observed annual maxima; solid lines and shaded areas represent, respectively, median and 90% confidence interval across the stations; dashed blue lines show the median for the case of GP tail model estimated from the synthetic Weibull tails (GP*); shaded grey areas in (a) and (b) quantify the stochastic uncertainty due to the available data record in presence of non-asymptotic Weibull tails. (c) Difference between L-skewness (purple) and L-kurtosis (green) derived from GP and Weibull tails (solid lines for the median, shaded areas for the 90% confidence interval) and from GP tails and GP* estimated from the synthetic Weibull tails (dashed, only the median is shown).

*This behavior is expected because, as we state in the paper, "The natural interpretation of these results would be that the Weibull distribution is a good model $F_Z$ for the parent process Z (positive rainfall or rainfall over low/moderate thresholds) confirming previous results reported in the literature, while GP model works well for exceedances over high thresholds (as postulated by EVT), and does not work well (as expected) for low/moderate thresholds, that is, outside its range of validity."*

The poor performance of positive rainfall with GP behavior up to moderately high threshold is not a limitation of GP, but an effect of using GP for "the body of the distribution". These diagrams (and the overall results reported by Marra et al.) do not call for a renewed consideration of nonasymptotic statistics (NA models of BM) for the description of extremes. They just translate in the following conclusions: *"(i) use GEV if only BM are available (e.g., AM from hydrologic reports), and (ii) use $F_Z$ (e.g., (compound) Weibull) if you have information on Z, which can be either the process of all positive rainfall or rainfall over arbitrary low/moderate thresholds if the latter is deemed easier to fit. In the latter case, calculate the T-year return levels as the (1− μ/T) · 100% quantiles of $F_Z$, where μ is the (mean) inter-arrival time (in years) between two observations of Z (e.g., Serinaldi, 2015; Volpi et al., 2019).*

*Such a plain reasoning highlights that there is no need to build an additional distribution of BM (i.e., SMEV, MEV or whatever else), in the same way we do not need to define the GEV distribution of AM once we already inferred a GP model of POT."*

Once (compound) Weibull (or anything else) is identified as an acceptable model for all positive rainfall or rainfall over arbitrary low/moderate thresholds, the behavior of any quantile (extreme or not) is completely described by this distribution. We do not need any other asymptotic or NA models of whatever surrogate process.

9. The manuscript presents numerous self-citations.

*Response* See our reply on CC1.

10.Incidentally, as a user of NA methods, I never claimed they are 'superlative'. They are as good as other models are: they offer advantages in some situations and disadvantages in others.

*Response* Thanks for this comment. We understand the Reviewer's point and in fact we never stated that the Reviewer claimed that NA models are superlative.

In our paper (and several previous papers of ours), we only criticize methods and/or other papers, not their Authors. However, we also understand that sometimes researchers tend to "fall in love"

and/or identify themselves with models and methods that they use and promote, especially if they do that for a long time.

We are confident that the reader can distinguish the different roles played by tone, style, and content in a written text, and therefore the intended meaning of the question "*Superlative or superfluous?*" in the paper title.

---

## Author Comment (AC3)

**Non-asymptotic distributions of water extremes: Superlative or superfluous?**

**By F. Serinaldi, F. Lombardo, C.G. Kilsby**

**Submitted to *HESS***

*MS-NR: hess-2023-234*
* * *
**Reply on CC1 (S. Han's report)**

(Note: In the text below, Referees' comments were copied verbatim in **black**.)

***Response*** We thank the Reviewer for the interesting feedback. In the following, we provide responses in **blue**.

I think that this manuscript is very dispersive. I suggest to insert (in the first part) at least one flow chart and one figure, in order to facilitate the understanding of all the steps for a common reader.

This is the difference between a very good scientific paper and a common one; in the latter case there could be the risk that only the authors and a very small set of readers can deeply understand the work!

Font size of Figures 2 and 3 seems very small! A very good dissemination of results also requests suitable figures. I suggest to enlarge the dimensions and (mainly for Figure 3) to create two figures, aimed at a better visualization (and understanding) of the plots.

Figures 4, 5 and 6: is logarithmic the scale of the vertical axes? I suppose it, but it is better if Author specify it along the text (or in the captions). This is always for a clear presentation of the results to all the readers in the scientific community.

Sections 5 and 6: my opinion about the "dispersion" is also confirmed by the presence of mathematical formulas in the second part of this manuscript. Indeed, the whole methodology description should be placed in the first part of a scientific paper, while the second part should be only dedicated to the discussion of the results.

Overall, this manuscript seems to suffer from two issues:

1. a not so clear (for all) presentation of the methodology;

2. self-referentiality: in the references part I counted twenty papers of Serinaldi, and this seems not so elegant in the scientific community…

Sincerely

***Response*** Again thanks for your time spent reading our manuscript. We will surely improve figures' font size and presentation in the revised version.

In the meantime, we would like to provide a response about the two key points raised by the Discussant:

- "Dispersion" (paper structure and materials' organization): As clearly stated in the introduction, our paper falls in the class of neutral validation papers, implying the analysis of the foundations of a given methodology. This requires an investigation of each stage of the considered methodology, bearing in mind that the conclusions/remarks/criticisms about each stage have consequences on the next one. Therefore, this type of papers should be read 'front-to-back' as an evolving story, and formulas are introduced when they are needed to discuss each specific point (other examples of this approach can be found in our previous papers cited in the text and references therein).
  In this respect, we purposedly introduce equations concerning the synthesis of multi-model ensembles in Section 5.3.1 so that the reader could benefit from fresh exposure to the concepts required to follow the discussion in the subsequent Section 5.3.2. Reporting those equations elsewhere (e.g., after Section 2 or in an appendix) would not allow a precise understanding of the inconsistencies discussed in Section 5.3.2. This understanding is of paramount importance for this type of papers, as their aim is precisely to show that neglecting theoretical aspects has direct/immediate (negative) consequences on the interpretation of numerical analysis (from real-world or synthetic datasets).
  We are sorry that the Discussant felt uncomfortable with the paper structure, which, however, cannot be of the type "methods-data-application-results", characterizing common papers. Papers' structure is not a dogma or an untouchable constraint. It generally varies according to journals, disciplines, and topics. Material organization is just the way we use to reach the aim, that is, communicating a message. Therefore, it should be chosen and adapted according to the aim and nature of the message we want to deliver, not vice versa.
  Moreover, the nature and structure of the paper are clearly stated in the introduction (L15-57 and L57-66, respectively). Therefore, the readers can be aware of the kind of paper they are going to read.
  Furthermore, our paper discusses technical inconsistencies of existing methods. This means that it explicitly assumes that the interested reader is familiar with the discussed methodologies.
  To summarize, the paper was purposely organized according to its nature and aim.
  In contrast to what stated by the Discussant, we think that the difference between 'a very good scientific paper and a common one' is not the materials' organization, but the rigour of the arguments. Presentation should adapt and follow accordingly.

- "Self-referentiality": This is an interesting topic; we are aware about the use, abuse, and misuse of self-references in a large amount of literature over the last twenty years or so. However, we just would like to provide the Discussant with the following food for thought:
  - We cite 20 papers of ours. However, the bibliography includes 107 references, and we are planning to add a few more (from other Authors) in the revised version. Thus, our papers cover less than the 19% of citations.
  - More importantly, references have (or should have) a purpose; thus, the actual point is not how many works of ours we cite, but if the references are appropriate and justified to support our statements. Constructive remarks should be: "these references do not deal with the topic or are inconsistent with the statements, please remove them", or "this topic is not introduced by these references for the first time, please use original references ('this' and 'that')", or again "this is a general concept presented in whatever handbook. Please, avoid citing yourself and use well-established references such as 'this' and 'that'".

Thus, the point is not "elegance" or "quantity" (which are always subjective and relative), but "how" and "why" we use citations. In this respect, we cite our own papers throughout the manuscript in two specific circumstances: (i) when we talk about technicalities concerning NA models because, as far as we know, Serinaldi et al. (2020) is the only paper providing a general/unified picture of NA models and a discussion of their unique/common nature based on the theory of order statistics; and (ii) when we talk about statistical inconsistencies widespread in the hydro-climatic literature and neutral validations. Also in the latter case, as far as we know, there are not many papers attempting independent validations of existing (but questionable) methods in hydro-climatology. We would be happy if more researchers were devoted to this activity, as happens in other disciplines, such as physics and medicine. Unfortunately, we believe this is not the case, and new methods are (too) often applied without the necessary validation of their theoretical foundations. Confusing iterative application with proper independent validation is detrimental for scientific progress, and this concept is one the main messages conveyed by our paper and previous papers of ours cited in the text.

Therefore, focusing on "quality", if the Discussant spotted inappropriate and/or missing citations we would be happy to remove incorrect references and add relevant ones that we may have missed.

That said, we report below our personal policy about referencing/citing our own work:

1) When we write a paper, we mention previous papers of ours only if we think that they can strengthen our message about very specific topics/issues that cannot be found elsewhere with similar flavor/point of view.

2)  We avoid citing our own papers when referring to general topics (e.g., EVT theory, order statistics, etc.). There is a huge amount of good books that can (and should) be cited, as they can absolutely be trusted.

3)  Our bibliographies are always quite extensive, and our works are always just a small fraction of the total number of cited references.

4)  When we act as reviewers and editors of other manuscripts, we mention our own works only to support "rejection", which means that the suggested references are expected not to be cited anywhere. Hardly ever, we suggest our own papers in review/editor reports implying "acceptance/revision". In those very rare cases, no more than 2-3 titles are suggested, and they are usually within a wider list of references from other Authors (thus, just a fraction of the total suggested references).

We hope that the above remarks clarified the rationale behind the paper structure/organization and its relationship with the paper message/aim, as well as our ethical policy concerning the use of references/citations and self-advertising methods.

---

## Author Response (AR1)

**Non-asymptotic distributions of water extremes: much ado about what?**

**Previously: "*Non-asymptotic distributions of water extremes: superlative or superflous!*"**

**By F. Serinaldi, F. Lombardo, C.G. Kilsby**

**Submitted to *HESS***

***MS-NR: hess-2023-234***
* * *
**REPLY ON EDITOR'S REPORT**

(Note: In the text below, Editor's comments were copied verbatim in **black**, whereas responses and changes in the text are in **blue**.)

Thanks to the authors and reviewers for detailed and thoughtful comments. While the paper primarily focusses on demonstrating the "redundancy and practical uselessness" of the Non-Asymptotic models of Block Maxima, however, in the conclusions section (lines 685-695), the authors adopt a more moderate and positive tone about the criticized approach. This is somehow contrasting with most of the paper's tone. The second referee (RC2 Report) also expressed a similar opinion about this topic.

***Response***
Dear Editor,

Thanks for giving us the opportunity to revise the manuscript and reply to Reviewers.
We understand that the tone of our paper is the only significant criticism to our work. In fact, all reviews we received acknowledged that our paper is technically correct, which is extremely encouraging for us.
Nevertheless, concerning the supposed contrast between the content reported in the text and the sentence extrapolated from the conclusions, we already discussed this issue in depth in our reply on RC2, which we report for convenience below:

"We kindly invite the Reviewer to read our paper more carefully. Indeed, it seems here that the Reviewer has extrapolated the meaning of some incomplete sentences out of their context giving rise to misunderstandings and wrong interpretations of the Authors' statements.
Actually, highlighting the difference between theory and practice is the very aim of our paper. Indeed, throughout the text, we always and purposely use periphrases like "*little usefulness for*

*practical* applications", "*usefulness of NA models in* **practical** *applications*", "*the problems concerning the use of NA models of BM for practical applications*", "*call into question the* **practical** *use and usefulness of NA*", etc., etc., whereas L685-694 refer to usefulness of NA models in *theoretical* context, as should be obvious to everyone.

The Reviewer is pointed to the distinction between practical and theoretical usage of NA models of BM, which is anticipated in L. 174-176, where we clearly state "*This explains why NA have not received much attention and why the recently proposed compound NA models are of* little **practical** *usefulness, if any. Their usefulness is mainly* **theoretical***, as they help explain the inherent differences between parent processes Z and BM processes Y, thus avoiding misconceptions and misinterpretation of different model outputs (see Serinaldi et al., 2020b).*"

Thus, it is quite evident that there is no contradiction at all.

Furthermore, Reviewer's remark does not consider the key point, that is, the argument supporting our conclusion about the practical uselessness of NA models of BM: "*NA models of BM imply the preliminary definition of their conditional parent distributions, which explicitly appears in their expression. However, when such conditional parent distributions are known or estimated also the unconditional parent distribution is readily available, and the corresponding NA distribution of BM is no longer needed, as it is just an approximation of the upper tail of the parent*"."

Therefore, we believe that "contradiction" may arise from (i) superficial reading (possibly neglecting entire sections), (ii) improper extrapolation of sentences out of the context, and/or (iii) biased reading. Anyway, in the revised paper we implemented some modification according to the editor suggestions.

Providing constructive criticism in a positive manner would enhance the paper's contribution to the scientific community in general. I recommend maintaining a positive tone throughout and avoiding terms as "useless" or "superfluous". Additionally consider a subtle title change to balance appeal and accuracy. While I understand that a more provocative title might capture a wider audience, I would suggest a subtle title change avoiding those terms.

*Response*
We understand Editor's point of view; therefore, we changed the paper title accordingly and eliminated the terms "superfluous" and "useless" throughout the revised text.

Regarding your statement in the RC1 document, specifically: "Under iid assumption, order statistics have a binomial distribution (which is equivalent to a beta distribution)". Please revisit these statements during your revised manuscript preparation and check for correctness.

*Response*
Thanks. We have clarified this point in the revised text as follows "*and the latter is described by a generalized beta distribution (see Eq. 1 as well as Eugene et al., 2002; Tahir and Cordeiro, 2016)*".

In fact, as mentioned in our reply on RC1:

"*Under i.i.d. assumption, order statistics have a binomial distribution (which is equivalent to a beta) in the sense described by Equation 1 (see also David and Nagarajah 2003, pp.9-10). In other words, the distribution of the order statistics is a beta distribution of the variable $F_Z(Z)$, or equivalently a so-called beta-extended distribution of Z (Eugene et al. 2002), which is also known as generalized beta-*

*G distribution, where "G" denotes generalized classes, such as exponentiated-G or Kumaraswamy-G (e.g., Tahir and Cordeiro, 2016)."*

The RC2 report extensively discusses the MCMC concept and the Bayesian approach. I recommend removing the example from section 5.3.1 that involves Monte Carlo methods within the Bayesian framework (i.e. MCMC). Instead consider providing an example more relevant to your actual research since the Bayesian approach is not utilized here.

**Response**
Thanks, we removed references to the Bayesian approach in the revised paper.

The RC2 discussion about the connection between physical processes and statistics is intriguing. However, it might be less relevant than considering the impact of the autocorrelation in the NA BM estimation methods described in section 5.2.

**Response**
To give some more food for thought about this issue, in the following we quote a recent miscellaneous work by Prof. Demetris Koutsoyiannis (2024; https://www.itia.ntua.gr/en/docinfo/2468/), who thoroughly addressed the distinction among real-world observations, models, and model outputs, as well as a vision of the relationship between physics and statistics that dates back to the 19th century:

*"… when applied to physical problems, statistical methods become parts of physics. Clockwise physics, without using probability and statistics, has been conventional wisdom for a couple of centuries but has proved to be weak and inadequate. Hence, stochastics has long ago been incorporated into physics. This occurred one century and a half ago, but admittedly, many of us… are not updated on this fact yet and continue to contrast physics and statistics. Therefore, I am providing the following information in bulleted form (along with my apology for being didactic):*

*• Statistical physics (cf. Boltzmann, Gibbs, Planck) used the probabilistic concept of entropy (which is nothing other than a quantified measure of uncertainty defined within the probability theory) to explain fundamental physical laws (most notably the Second Law of thermodynamics), thus leading to a new understanding of natural behaviors and to powerful predictions of macroscopic phenomena. Atmospheric processes are explained by statistical physics in all respects (thermodynamic equilibrium, blackbody radiation, transport processes)*

*• Quantum theory (cf. Heisenberg) has emphasized the intrinsic character of uncertainty and the necessity of probability in the description of nature.*

*• Developments in numerical mathematics for applications in physics (cf. Metropolis) highlighted the effectiveness of stochastic methods in solving physical problems that are even purely deterministic, such as numerical integration in high-dimensional spaces and global optimization of non-convex functions (where stochastic techniques, e.g., stochastics-based evolutionary algorithms and simulated annealing, are in effect the only feasible solution in complex problems that involve many local optima).*
*This extends even beyond physics. Thus,*

- *Genetics (cf. Mendel) and evolutionary biology have emphasized the importance of stochasticity (e.g., in gametes fusion, selection and mutation procedures, and environmental changes) as a driver of evolution.*

- *Developments in mathematical logic, and particularly Gödel's incompleteness theorem, challenged the almightiness of deduction (inference by mathematical proof). This necessitates the use of induction in physical problems, whose theoretical basis is offered by the field of stochastics".*

Thus, Reviewer#2's statement "*One notable example is the connection between physical processes and statistics, which can only exist in a NA model, given that real world physics is not asymptotic. It is my believe that physical processes should direct the statistics we use. The physics of the processes we are dealing with is not asymptotic*" makes little sense because:

- NA models are just tools devised for very specific finite-size-sample problems.
- Physics, statistical physics, quantum physics, etc., are neither asymptotic nor non-asymptotic; these properties only concern how models specialize for the problem, data, and spatio-temporal scales at hand.

A clearer statement about the role of physics and statistics in hydrological modeling is provided by Montanari and Koutsoyiannis (2014; https://doi.org/10.1002/2014WR016092), which we report verbatim for convenience:

"*We believe that there is a widespread misconception in the hydrologic community, related to the use of process-based versus statistical models. The prevailing view is that process-based deterministic models are deductive means that take advantage of the available knowledge of the process dynamics, while statistical models are inductive and therefore are useful when the above knowledge is limited. We believe that this view is inconsistent. In complex hydrological systems, both deterministic and stochastic models are necessarily inductive (as they rely on fitting on data), while any deductive component in a deterministic model can be conveyed also in a stochastic model [Montanari and Koutsoyiannis, 2012]. The actual difference between deterministic and statistical models is just that the former establish a precise relationship between input (including initial and boundary conditions) and output (including system state), while the latter examines the probabilities of events (or time evolution thereof) by admitting that randomness, and therefore uncertainty, is inescapable. A statistical or stochastic model is just not deterministic: it can be physically based, it can represent spatial and time variability and can take full advantage of the knowledge of the system. Because of this, stochastic models with an increasing content of physical reasoning have been gaining increasing attention over the last decades. In order to identify the appropriate model to use, one should simply decide whether one wants to represent the inherent randomness affecting hydrological processes, and whether or not one wants to take uncertainty into account. There is no doubt that process-based models are the most appropriate solution for solving many water related problems, but we do not see any reason not to formulate them in a stochastic context. In our opinion, stochastic-process-based models are the way forward to bridge the gap between physically-based models without statistics and statistical models without physics. There has been a lot of applications in hydrology that clarified the potential of stochastic process-based models*".

Building stochastic process-based models only concerns incorporating uncertainty and has nothing to do with asymptotic or non-asymptotic models. For example, are stochastic differential equations (SDEs) asymptotic or non-asymptotic? SDEs are stochastic process-based models including

deterministic terms (describing the dominant evolution some dynamical system) and stochastic terms accounting fluctuations that cannot be described by the deterministic part.

In lieu of the two reviewers' reports and community comments I suggest a major revision of your manuscript along the lines suggested by the reviewers and my own recommendations.

*Response*
We thank once again the handling Editor and Reviewers for the time devoted to our paper. Concerning the specific remarks:

- As per Reviewer#1's request, we revised the notation and some unclear sentences. Please, note that HESS provides language editing at proofreading stage that will fix residual problems.
- Concerning Reviewer#2, he explicitly stated that there are no technical errors (apart from typos in some formulas spotted by Reviewer#1). On the other hand, our detailed responses clarify that Reviewer#2's remarks are just part of a classical scientific debate between a researcher who supports NA models for BM and researchers that wrote a technically correct manuscript that criticizes such models. In fact, recognizing that our results are technically correct implies that our work has some value for the hydrological community, and it possibly deserves publication in HESS.
- Concerning the Discussant's comments, we updated the figures to make them more readable.
- As far as the remarks about style, length, and presentation are concerned, we revised the text according to the suggestions of the Editor and Reviewers bearing in mind the importance of repeating some crucial piece of information to ensure reception by readers of various background.
  We also noted that the Reviewers mostly focused on Section 5.3.1. However, our work reports theoretical and conceptual reasoning that justify our point of view as well as data analysis (Section 4) and extensive Monte Carlo simulations (Sections 5.1. and 5.2). These simulations reproduce and extend the numerical experiments reported by Reviewer#2 in one of his papers. Moreover, some Reviewers' comments about Section 5.3.1 seem not to account for its premises and consequences reported in Section 5.3.2.
  Therefore, we stress again our belief that our work has some value for the hydrological community, and it possibly deserves publication in HESS.